# PARETO OPTIMALITY IN NO-HARM FAIRNESS

## ABSTRACT

Common fairness definitions in machine learning focus on balancing various notions of disparity and utility. In this work we study fairness in the context of risk disparity among sub-populations. We introduce the framework of *Pareto-optimal fairness*, where the goal of reducing risk disparity gaps is secondary only to the principle of not doing unnecessary harm, a concept that is especially applicable to high-stakes domains such as healthcare. We provide analysis and methodology to obtain maximally-fair no-unnecessary-harm classifiers on finite datasets. We argue that even in domains where fairness at cost is required, no-unnecesary-harm fairness can prove to be the optimal first step. This same methodology can also be applied to any unbalanced classification task, where we want to dynamically equalize the misclassification risks across outcomes without degrading overall performance any more than strictly necessary. We test the proposed methodology on real case-studies of predicting income, ICU patient mortality, classifying skin lesions from images, and assessing credit risk, demonstrating how the proposed framework compares favorably to other traditional approaches.

## 1 INTRODUCTION

Machine learning algorithms play an important role in decision making in society. When these algorithms are used to make high-impact decisions such as hiring, credit-lending, predicting mortality for intensive care unit patients, or classifying benign/malign skin lesions, it is paramount to guarantee that these decisions are both accurate and unbiased with respect to sensitive attributes such as gender or ethnicity. A model that is trained naively may not have these properties by default; see, for example Barocas & Selbst (2016).

In these critical applications, it is desirable to impose some fairness criteria. Much of the fairness in machine learning literature attempts to produce algorithms that satisfy Demographic Parity, which aims to make algorithm's predictions independent of the sensitive populations (Louizos et al. (2015); Zemel et al. (2013); Feldman et al. (2015)); or Equality of Odds or Equality of Opportunity, which aims to produce predictions that are independent of the sensitive attributes given the ground truth (Hardt et al. (2016); Woodworth et al. (2017)). Notions of Individual Fairness have also been advanced (Dwork et al. (2012); Joseph et al. (2016); Zemel et al. (2013)). These notions of fairness can be appropriate in many scenarios, but in domains where quality of service is paramount, such as healthcare, we argue that it is necessary to strive for models that are as close to fair as possible without introducing any unnecessary harm to any subgroup (Ustun et al. (2019)). Even if the overall fairness goal is a potentially harmful, zero-gap classifier, pursuing first a Pareto-fair classifier and later applying other harmful methodologies ensures that all possible non-harmful trade-offs are covered before explicitly degrading performance, therefore minimal harm is introduced to the decision.

In this work we make use of the concept of Pareto optimality to measure and analyze discrimination (unfairness) in terms of the difference in predictive risks across sub-populations defined by our sensitive attributes, a fairness metric that has been explored in other recent works such as Calders & Verwer (2010); Dwork et al. (2012); Feldman et al. (2015); Chen et al. (2018); Ustun et al. (2019). We examine the subset of models from our hypothesis class that have the best trade-offs between sub-population risks, and select from this set the one with the smallest risk disparity gap. This is in direct contrast to common post-hoc correction methods like the ones proposed in Hardt et al. (2016); Woodworth et al. (2017), where noise is potentially added to the decisions of the best performing sub-population. While this latter type of approach diminishes the risk-disparity gap, it does so by degrading performance on advantaged groups, the previously disadvantaged groups do not directly benefit from this treatment. Since our proposed methodology does not require test-time

access to sensitive attributes, and can be applied to any standard classification or regression task, it can also be used to reduce risk disparity between outcomes, acting as an adaptive risk equalization loss compatible with unbalanced classification scenarios.

**Main Contributions.** We formalize the notion of no-harm risk fairness[1] using Pareto optimality (Mas-Colell et al. (1995)), a state of resource allocations from which it is impossible to reallocate without making one subgroup worse. We show that finding a Pareto-fair classifier is equivalent to finding a model in our hypothesis class that belongs to the Pareto front (the set of all Pareto optimal allocations) with respect to the sub-population risks with the smallest possible risk disparity. This general notion is already amenable to non-binary sensitive attributes. We analyze the fairness performance trade-offs that can be expected from different approaches with an illustrative example. We provide a concrete algorithm that promotes fair solutions belonging to the Pareto front; this algorithm can be applied to any standard classifier or regression task that can be trained using (Stochastic) Gradient Descent. We show that if the goal is to obtain a zero-gap classifier, first recovering the fairest Pareto optimal solution and then adding harmful post-hoc corrections ensures the lowest risk levels across all subgroups. We demonstrate how our methodology performs on synthetic and real tasks such as inferring income status in the Adult dataset Dua & Graff (2017a) irrespective of their ethnicity or gender, predicting ICU mortality rates in the MIMIC-III dataset from hospital notes Johnson et al. (2016), classifying skin lesions in the HAM10000 dataset Tschandl et al. (2018), and assessing credit risk on the German Credit dataset Dua & Graff (2017b).

## 2 RELATED WORK

There is an extensive body of work on fairness in machine learning. Following Friedler et al. (2019), we compare our methodology against the works of Feldman et al. (2015); Kamishima et al. (2012); Zafar et al. (2017). Our method shares conceptual similarities with Zafar et al. (2017); Woodworth et al. (2017); Agarwal et al. (2018), with differences on how we define our fairness objective and adapt it to work with standard neural networks. Although optimality is often discussed in the fairness literature, it is usually in the context of utility-discrimination tradeoffs. To the best of our knowledge, this is the first work to discuss optimality with respect to subgroup risks on a unified classifier, a distinction that disallows extreme performance degradation in the pursuit of fairness.

The work presented in Hashimoto et al. (2018) discusses decoupled classifiers as a way of minimizing group-risk disparity, but simultaneously cautions against this methodology when presented with insufficiently large datasets. The works of Chen et al. (2018); Ustun et al. (2019) also empirically report the disadvantages of decoupled classifiers as a way to mitigate risk disparity. Here we argue for the use of a single classifier because it allows transfer learning between diverse sub-populations. We do not need access to the sensitive attribute during test time, but in cases where this is possible, we instead choose to incorporate it as part of our observation features.

The work of Chen et al. (2018) uses the unified bias-variance decomposition advanced in Domingos (2000) to identify that noise levels across different sub-populations may differ, making perfect fairness parity impossible without explicitly degrading performance on one subclass. Their methodology attempts to bridge the disparity gap by collecting additional samples from high-risk sub-populations. Here we modify our classifier loss to bridge the disparity gap without inducing unnecessary harm, which could prove to be synergistic with their methodology.

## 3 PROBLEM STATEMENT

Consider we have access to a dataset $\mathcal{D} = \{(x_i, y_i, a_i)\}_{i=1}^n$ containing $n$ independent triplet samples drawn from a joint distribution $(x_i, y_i, a_i) \sim P(X, Y, A)$ where $x_i \in \mathcal{X}$ are our input features (e.g., images, tabular data, etc.), $y_i \in \mathcal{Y}$ is our target variable, and $a_i \in \mathcal{A}$ indicates group membership or sensitive status (e.g., ethnicity, gender); our input features $X$ may or may not explicitly contain $A$.

Let $h \in \mathcal{H}$ be a classifier from a compact hypothesis class $\mathcal{H}$ trained to infer $y$ from $x$, $h : \mathcal{X} \to \mathcal{Y}$; and a loss function $\ell : \mathcal{Y} \times \mathcal{Y} \to \mathbb{R}$. We define the class-specific risk of classifier $h$ on subgroup $a$ as $R_a(h) = \mathbb{E}_{X,Y|A=a}[\ell(h(X), Y)]$. The risk discrimination gap between two subgroups $a, a' \in \mathcal{A}$ is

---

[1]In this paper we use "no-harm" to mean "no-unnecessary-harm," in other words, the system doesn't degrade performance on any class unless it is strictly necessary to improve performance in a disadvantaged class.

measured as $\Gamma_{a,a'}(h) = |R_a(h) - R_{a'}(h)|$, and we define the pairwise discrimination gap vector as $\vec{\Gamma}_\mathcal{A}(h) = \{\Gamma_{a,a'}(h)\}_{a,a'\in\mathcal{A}}$. Our goal is to obtain a classifier $h \in \mathcal{H}$ that minimizes this gap without causing unnecessary harm to any particular group in $\mathcal{A}$. To formalize this notion, we define:

**Definition 3.1.** Dominant risk vector: A vector $\boldsymbol{r}' \in \mathbb{R}^k$ is said to dominate $\boldsymbol{r} \in \mathbb{R}^k$, noted as $\boldsymbol{r} \succ \boldsymbol{r}'$, if $\boldsymbol{r}_i \geq \boldsymbol{r}'_i, \forall i = 1,...,k$ and $\exists j : \boldsymbol{r}_j > \boldsymbol{r}'_j$ (i.e., strict inequality on at least one component).

**Definition 3.2.** Dominant risk classifier: Classifier $h'$ is said to dominate $h$, noted as $h \succ h'$, if the risks vector $\boldsymbol{r}' = \{R_a(h')\}_{a=1}^{|\mathcal{A}|}$ dominates $\boldsymbol{r} = \{R_a(h)\}_{a=1}^{|\mathcal{A}|}$.

**Definition 3.3.** Pareto front: We define the *Pareto front* as $\mathcal{P}(\mathcal{H}, \mathcal{A}) = \{h \in \mathcal{H} : \nexists h' \in \mathcal{H} \mid h \succ h'\}$. This means that there is no other classifier in $\mathcal{H}$ that is at least as good in all risks and strictly better in at least one of them. It is the set of classifiers such that improving one group's risk comes at the cost of increasing other's.

The Pareto front defines the best achievable trade-offs between population risks $R_a(h)$. This definition is already suited for classification and regression tasks where the sensitive attributes are categorical. Constraining the classifier to be in the Pareto front disallows laziness, there exists no other classifier in the hypothesis class $\mathcal{H}$ that is at least as good on all class-specific risks and strictly better in one of them. As shown in Chen et al. (2018); Domingos (2000), the risk can be decomposed in bias, variance and noise for some loss functions, where the noise is the smallest achievable risk for infinitely large datasets (Bayes-optimal risk). If the noise differs between sensitive groups, zero-discrimination (perfect fairness) can only be achieved by introducing bias or variance.

Literature on fairness has focused on putting constraints on the norm of discrimination gaps (Zafar et al. (2017; 2015); Creager et al. (2019); Woodworth et al. (2017)). We follow a similar criteria in Definition 3.4 and define the Pareto-fair classifier as the classifier in the Pareto front that minimizes $||\vec{\Gamma}_\mathcal{A}(h)||_\infty$ (the maximum risk discrimination gap). Note that one could alternatively choose to find the Pareto classifier that minimizes the maximum subgroup risk.

**Definition 3.4.** Pareto-fair classifier and Pareto-fair vector: A classifier $h^*$ is an optimal Pareto-fair classifier if it minimizes the discrimination gap among all Pareto front classifiers, $h^* = \arg\min_{h\in\mathcal{P}(\mathcal{H},\mathcal{A})} ||\vec{\Gamma}_\mathcal{A}(h)||_\infty$. The Pareto-fair vector $\boldsymbol{r}^* \in \mathbb{R}^{|\mathcal{A}|}$ is defined as $\boldsymbol{r}^* = \{R_a(h^*)\}_{a\in\mathcal{A}}$.

Even when perfect equality of risk is desirable, Pareto classifiers still serve as useful intermediaries. To this end, Lemma 3.1, shows that applying a mechanism for reaching equality of risk on a dominated classifier $h \in \mathcal{H}$ leads to equal or worse risks than applying it to a Pareto classifier $h_p \in \mathcal{P}(\mathcal{H}, \mathcal{A})$ that dominates $h$.

**Lemma 3.1.** *If $h \notin \mathcal{P}(\mathcal{H}, \mathcal{A}) \to \exists h_p \in \mathcal{P}(\mathcal{H}, \mathcal{A}) : h_p \succ h \wedge R_a(h_p^{ER}) \leq R_a(h^{ER}) \forall a$, with $h^{ER}$ an equal-risk classifier* : $R_a(h^{ER}) = \max_{a'\in\mathcal{A}} R_{a'}(h), \forall a$ *and* $h_p^{ER} : R_a(h_p^{ER}) = \max_{a'\in\mathcal{A}} R_{a'}(h_p)$.

To exemplify these notions graphically, Figure 1 shows a scenario with binary sensitive attributes $a$ and binary output variable $y$ where none of the Pareto front classifiers achieve equality of risk. Here the noise level differs between subgroups, and the Pareto-fair vector $\boldsymbol{r}^*$ is not achieved by either a Naive classifier (minimizes expected global risk), or a classifier where subgroups are re-sampled to appear with equal probability (rebalanced Naive classifier). Note that the amount of performance degradation required to enforce perfect fairness starting from the Naive classifier is higher than when starting from the Pareto-fair vector.

Our objective is to find the Pareto-fair classifier $h^*$ as in Definition 3.4. In particular, we will give regularity conditions on the hypothesis class $\mathcal{H}$ and the loss function $\ell(h(X), Y)$. It can be shown that for a sufficiently rich class of hypothesis functions $\mathcal{H}$ and for risk functions $R_a(h)$ that are convex with respect to $h$, the *space of risk vectors is convex* (see Geoffrion (1968); Koski (1985); Miettinen (2012)). Under these conditions, we can find an auxiliary loss function $\phi : \mathbb{R}^{|\mathcal{A}|} \to \mathbb{R}$ defined in terms of the subgroup risks $\{R_a(h)\}_{a\in\mathcal{A}}$ (denoted from here on as $\boldsymbol{r} \in \mathbb{R}^{|\mathcal{A}|}$) that has a global minima on $\boldsymbol{r}^*$.

To prove the existence of loss function $\phi$ with the desired global minimum $\boldsymbol{r}^*$, it is convenient to think of the convex set of risk vectors as the intersection of a convex Pareto set (defined by a convex Pareto front and all risks that are dominated by it), and an additional convex set $\Omega$. We further require both this Pareto set and $\Omega$ to be smooth, so that we can apply standard tools from smooth convex optimization and prove that for every risk vector $\boldsymbol{r}'$ in the Pareto front there exists a function $\phi$ that has $\boldsymbol{r}'$ as its global optima. This is formalized in Lemma 3.3.

**Definition 3.5.** Pareto field: A function $P : \Omega \to \mathbb{R}$ is a *Pareto field* over a convex set $\Omega \subset \mathbb{R}^k$ if $P \in \mathcal{C}^1$ is a continuously differentiable function such that $\nabla_i P(\boldsymbol{r}) > 0 \, \forall \boldsymbol{r} \in \Omega, \forall i = 1, \ldots, k$.

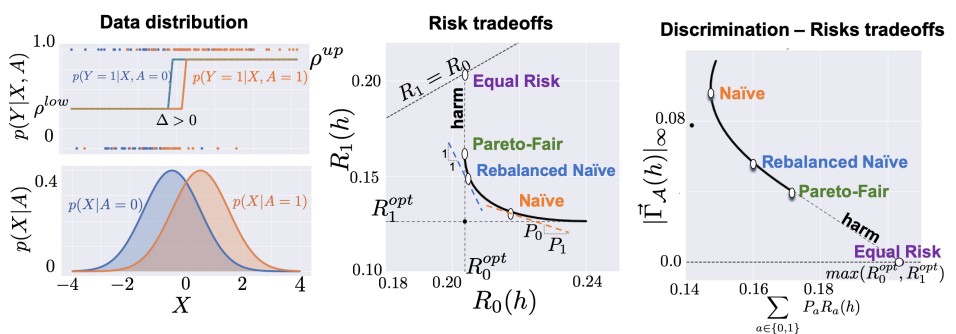

Figure 1: Example of achievable risk tradeoffs for a binary regression problem with two unbalanced sensitive groups $a = \{0, 1\}$ ($P_{a=1} > P_{a=0}$). Bottom left and top left figures show conditional distributions of observation variable $p_a(x)$, and target variable $p(y = 1|x, a)$ respectively. $p(y = 1|x, a)$ are simple piece-wise constant functions with levels $\rho^{low}$ and $\rho^{high}$. Middle figure shows the Pareto front defined by subgroup risks; we observe that noise levels differ across subgroups and perfect Pareto fairness is unattainable. Equality of Risk requires pure degradation of service for group $a = 1$. Both Naive and Rebalanced Naive classifiers do not attain the best possible no-harm classifier in this case. The proposed Pareto-fair point is shown in green, with its corresponding optimal Equal risk classifier shown in purple. The utopic point ($R_0^{opt}$, $R_1^{opt}$) can only be attained when the classifier also has access to the sensitive attribute ($h(X, A)$). Rightmost figure shows the trade-offs attainable between discrimination and mean risks, an alternative perspective to the risk trade-off figure.

**Lemma 3.2.** *Let* $\Omega \subset \mathbb{R}^k$ *be a convex set, and* $P : \Omega \rightarrow \mathbb{R}$ *a Pareto field. Then the set* $D = \{r \in \Omega : P(r) = 0\}$ *is a Pareto set in* $\Omega$*, and the set* $D^+ = \{r \in \Omega : P(r) > 0\}$ *is the set of dominated points, i.e.,* $D^+ = \{r \in \Omega : \exists r' \in D \mid r \succ r'\}$

**Lemma 3.3.** *Let* $\Omega \subset \mathbb{R}^k$ *be a convex set defined by* $\Omega = \{r \in \mathbb{R}^k : g^c(r) \geq 0 \forall c \in \{1, \ldots, C\}, g^c \text{ continuously differentiable}\}$; *let* $P : \Omega \rightarrow \mathbb{R}$ *be a convex Pareto field with corresponding proper Pareto set* $D = \{r \in \Omega : P(r) = 0\}$*. Let* $\hat{r} \in D$ *and* $\phi(r, \mu) = \sum_{i=1}^k r_i + \mu_i(r_i - c)^{+2}$*, with* $c < \hat{r}$*, where we denote* $(x)^{+2} = \max^2(x, 0)$*. There exists a set of vectors* $\hat{\mu} \succ 0$ *such that:*

$$\hat{r} = \arg\min_{r \in \mathbb{R}^k} \phi(r, \hat{\mu}) \quad s.t. : P(r) \geq 0, g^c(r) \geq 0 \forall c \in \{1, \ldots, C\}.$$

Where $C$ is the number of constraints that characterized $\Omega$. Proofs for all lemmas are given in the supplementary material, Section A.1. Lemma 3.3 motivates our loss function to take the form $\phi(r, \mu) = \sum_{i=1}^{|\mathcal{A}|} r_i + \mu_i(r_i - c)^{+2}$, with $|\mathcal{A}|$ the number of sensitive groups. The challenge is to find $\mu^*$ such that our Pareto-fair vector $r^*$ minimizes $\phi(r, \mu^*)$. In Section 4 we provide an algorithm to approximately recover the Pareto-fair classifier $h^*$ by adaptively searching for $\mu^*$ and optimizing a classifier on $\phi(\cdot, \mu)$ using tools from standard Stochastic Gradient Descent.

## 4 OPTIMIZATION METHODS

Recall that we wish to recover the Pareto-fair classifier $h^*$ within our hypothesis class. From Lemma 3.3, there exists a loss function of the form

$$
\begin{aligned}
\phi(r; \mu, c) &= \sum_{i=1}^{|\mathcal{A}|} r_i + \mu_i(r_i - c)^{+2}, \\
\phi(h; \mu, c) &= \sum_{a \in \mathcal{A}} R_a(h) + \mu_a(R_a(h) - c)^{+2},
\end{aligned}
\tag{1}
$$

such that $h^* = \arg\min_{h \in \mathcal{H}} \phi(h; \mu, c)$. Note that we can state the loss function directly on the risk vectors $r$, or implicitly on the classifier $h$. Since $R_a(h)$ is differentiable with respect to $h$, $\phi(h; \mu, c)$ can be directly minimized using gradient descent on $h$ for any choice of values $\mu, c$. Let $\mathcal{D}^{\text{Tr}} =$

$\{x, y, a\}_{i=1}^{N_T}$, $\mathcal{D}^{\text{Val}} = \{x, y, a\}_{i=1}^{N_V}$ be our training and validation datasets respectively. The proposed implementation of the Pareto-fair framework is formalized in Algorithm 1 where we specify how to update the penalty coefficients $\vec{\mu}, c$.

---

**Algorithm 1:** ParetoFairOptimization

---

**Given:** $h_\theta, \ell, \mathcal{D}^{\text{Tr}}, \mathcal{D}^{\text{Val}}, n_\mu, n_p, n_{max}, \gamma > 0, k \geq 1, \xi, \zeta \in (0, 1), \text{lr}, \text{B}$

$\boldsymbol{\mu} \leftarrow \mathbf{1}$, $\boldsymbol{\mu}^* \leftarrow \mathbf{1}$, $\mu_{\text{count}} \leftarrow 0$, $e_{\text{count}} \leftarrow 0$, $c \leftarrow 0$, $\Gamma^* \leftarrow \infty$, $h^* \leftarrow h_\theta$

**while** $e_{count} \leq n_{max}$ and $\mu_{count} \leq n_\mu$ **do**

    $\mu_{\text{count}} \leftarrow \mu_{\text{count}} + 1$, $e_{\text{count}} \leftarrow e_{\text{count}} + 1$

    $h_\theta, \boldsymbol{r}^{\text{Val}} \leftarrow \text{AdaptiveOptimize}(h_\theta, \ell, \boldsymbol{\mu}, c, \mathcal{D}^{\text{Tr}}, \mathcal{D}^{\text{Val}}, n_p, \text{lr}, \text{B})$ // Optimize current loss

        // Check that solution is Pareto efficient and reduces fairness gap

    **if** $||\vec{\Gamma}_{\mathcal{A}}(h)||_\infty < \Gamma^*$ and $\boldsymbol{r}^{Val}$ is not dominated by previous validation risks **then**

        $h^* \leftarrow h_\theta$, $\Gamma^* \leftarrow ||\vec{\Gamma}_{\mathcal{A}}(h)||_\infty$, $c_{\text{old}} \leftarrow c$, $c \leftarrow \min_a \frac{r_a^{\text{Val}}}{k}$

        $\boldsymbol{\mu}^* \leftarrow \boldsymbol{\mu} \cdot \frac{(\boldsymbol{r}^{\text{Val}} - c_{\text{old}})^+}{(\boldsymbol{r}^{\text{Val}} - c)^+}$, $a' \leftarrow \arg\max_a r_a^{\text{Val}}$, $\mu_{\text{count}} \leftarrow 0$

    **else**

        $\text{lr} \leftarrow \zeta \, \text{lr}$, $\boldsymbol{\mu} \leftarrow \boldsymbol{\mu}^*$, $\gamma \leftarrow \gamma\xi$, $h_\theta \leftarrow h^*$

    **end**

    $\mu_{a'} \leftarrow (1 + \gamma)\mu_{a'}$

**end**

    // Exit loop due to excessive iterations or no improvement in fairness

**Return:** $h^*$

---

We regularly check that reductions in the fairness gap generalize to the validation set; we additionally check if the trade-offs are in the non-dominated solution set (i.e., we have not observed a universally better classifier during training). Algorithm 2 (AdaptiveOptimize, shown in Section A.2) summarizes how we perform stochastic gradient descent steps with early stopping in-between $\boldsymbol{\mu}, c$ updates. Lemma A.5 (shown in supplementary material, Section A.1) shows that this algorithm is convergent for $|\mathcal{A}| = 2$ when the minimization step for a fixed $\boldsymbol{\mu}, c$ is performed exactly.

To conclude this section, we stress that the proposed framework is independent of the desired algorithm class $\mathcal{H}$ and loss function $\ell$; these are kept from the original application. The Pareto-fair classifier uses the same inputs as the Naive classifier, with parameters that have been optimized towards Pareto fairness. Code will be made available.

## 5 EXPERIMENTS AND RESULTS

We applied the methodology described in Section 4 to learn a Pareto-fair classifier (classifier in the group-risk Pareto front with the smallest risk disparity). We first validate our methodology on synthetic data with known Pareto-fair classifiers. Observations are drawn from a Gaussian mixture model where each sensitive attribute is encoded by a corresponding Gaussian mode, and target attributes are binary. We demonstrate our methodology on publicly available fairness datasets, and show how the risk disparity gaps are subsequently reduced. Where applicable, we compare our results against the methodologies proposed in Zafar et al. (2017); Kamishima et al. (2012); Hardt et al. (2016); Feldman et al. (2015).

### 5.1 SYNTHETIC DATA

We tested our approach on synthetic data where the observations are drawn from conditional Gaussian distributions $X|A = a \sim N(\mu_A, 1)$, the target variable $y$ is a conditional Bernoulli variable with distribution $Y|X = x, A = a \sim Ber\big(f_a(x)\big)$ with $f_a(x) = \rho_a^{low}\mathbb{1}[x \leq c_a] + \rho_a^{high}\mathbb{1}[x > c_a])$, and $|A| = 3$. We used Brier Score (BS, refer to Section A.4) as our loss function. Under these conditions, the Bayes-optimal classifier for each subgroup is $h(x) = f_a(x)$. The subgroup risk function for any classifier $h$ can be computed as $R_a(h) = \mathbb{E}_{X|A=a}[Y - h(X))^2] = \mathbb{E}_{X|A=a}[f_a(X)(1 - h(X))^2 + (1 - f_a(X))h(X)^2]$. Network details are given in the supplementary material, Section A.6. Figure 2 shows the analytically-derived Pareto-fair classifier, as well as the trade-off obtained by the proposed algorithm. We remark that the Pareto-fair classifier in this scenario cannot be achieved using linear logistic regression, something that can be seen from the functional form of the Bayes Optimal classifier derived in Section A.8 in supplementary material. From the center graph in Figure 2 we can see that the approximated Pareto-Fair risks (dashed

lines) are close to the theoretical best (solid lines).The achieved maximum risk discrimination is also close to the theoretical best. The algorithm is able to correctly trade risks from the two advantaged subgroups to reduce the risk of the worst performing subgroup.

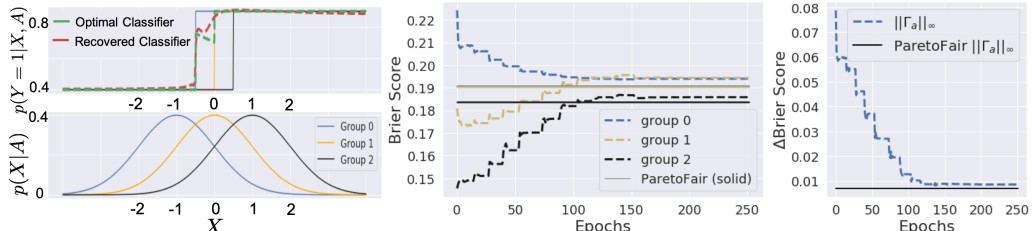

Figure 2: Synthetic data experiment. Bottom left figure shows the conditional distribution of the observation variables for each of the three subgroups, while upper left shows the distribution of the target variable conditioned on both the observation and sensitive attribute; it additionally shows the theoretical and empirical Pareto-fair classifier. Center figure shows the empirical validation risks (dashed lines) as a function of epochs during the optimization procedure, the theoretical Pareto-Fair risks are also shown (solid lines). Rightmost shows how the maximum discrimination shrinks with the number of epochs.

## 5.2 REAL DATASETS

We evaluate the performace of our algorithm on mortality prediction (MIMIC-III), skin lesion classification (HAM10000), income prediction (Adult), and credit lending (German). We also compared with several state of the art methods (Zafar et al. (2017); Kamishima et al. (2012); Hardt et al. (2016); Feldman et al. (2015)) and with a model trained to minimize the average risk (Naive) and one that undersamples the minority class (Naive Balanced or NaiveB). A description of these methods is provided in Section A.3 of Supplementary material. Metrics used for evaluation include accuracy (Acc), confidence (Conf), expected and maximum calibration error (ECE and MCE respectively), Brier score (BS) and cross-entropy (CE). Values for every metric are reported per sensitive attribute. The dataset average is reported as (sample mean), and the group-normalized average as (group mean). The best and worst performing groups are also shown as (group max) and (group min), and the worst case difference between subgroups is reported as (disc). For an in-depth description of the the metrics and datasets used for evaluation, refer to Sections A.4 and A.5 of Supplementary material. Here we present a comparison of the performance in terms of accuracy and ECE but an extensive comparison is available in additional results Section A.7 in supplementary material.

Similarly to Chen et al. (2018); Zafar et al. (2017); Hardt et al. (2016); Woodworth et al. (2017), we omit the sensitive attribute $a \in \mathcal{A}$ from our observation features. Our method trains a single classifier for the entire dataset to avoid needing test-time access to sensitive attributes whenever possible. All classifiers shown in this section are implemented using neural networks and/or logistic regression, we used either cross-entropy or Brier score as our training loss depending on the dataset. For details on the architecture and hyperparameters used on each dataset, refer to the supplementary material, Section A.6. We show how the proposed Pareto-fair approach produces well calibrated models that reduce group disparities across several metrics.

### 5.2.1 PREDICTING MORTALITY IN INTENSIVE CARE PATIENTS

Medical decisions in general and mortality prediction in particular are examples where notions of fairness among sub-populations are of paramount importance, and where ethical considerations make no-harm fairness a very attractive paradigm. To that end, we used clinical notes collected from adult ICU patients at the Beth Israel Deaconess Medical Center (MIMIC-III dataset) Johnson et al. (2016) to predict patient mortality. We study fairness with respect to age (adult or senior), ethnicity (white or nonwhite), and outcome (alive/deceased). This leads to a total of 8 sensitive groups. We included outcome (alive/deceased) as a sensitive sub-population criteria to demonstrate a case where sensitive attributes would not be available at test-time, and because in our experiments patients who ultimately passed away on ICU were under-served by a Naive classifier. Table 1 shows empirical accuracies and expected calibration errors of all tested methodologies. It is important to note how imbalanced these groups are by looking at the 'ratio' column. Here one can see that 56.7% of the samples correspond to the majority class (alive, white, senior) and only 0.4% to the minority

(a) Accuracy comparison

| type | ratio | Naive | NaiveB | ZafarCE | ParetoFair CE | ParetoFair BS |
|---|---|---|---|---|---|---|
| sample mean | - | **89.5±0.2%** | 78.9±0.7% | 78.0±0.7% | 74.3±0.8% | 75.7±1.1% |
| group mean | 12.5% | 61.8±1.5% | 77.3±1.3% | 69.4±1.3% | 76.2±0.5% | **77.5±0.7%** |
| group max | 56.7% | **99.2±0.3%** | 88.1±1.3% | 93.0±1.4% | 86.4±2.2% | 87.1±2.0% |
| group min | 0.4% | 18.1±2.5% | 65.7±1.2% | 43.7±3.9% | 67.9±2.5% | **69.9±2.2%** |
| disc | 56.3% | 81.1±2.5% | 22.5±2.5% | 49.3±4.4% | 18.5±4.2% | **17.1±2.6%** |

(b) Accuracy comparison after Hardt post-processing

| type | ratio | NaiveB+H | Zafar+H | ParetoFair CE+H | ParetoFair BS+H |
|---|---|---|---|---|---|
| sample mean | - | **75.1±1.8%** | 66.3±1.0% | 70.6±1.3% | 71.9±1.2% |
| group mean | 12.5% | 71.2±1.1% | 62.4±1.3% | 70.8±0.7% | **72.3±1.0%** |
| group max | 56.7% | **80.0±2.5%** | 70.0±2.6% | 77.6±4.1% | 79.9±5.6% |
| group min | 0.4% | 61.4±3.4% | 52.0±3.6% | 65.7±1.8% | **66.8±2.1%** |
| disc | 56.3% | 18.6±3.2% | 18.1±4.3% | **11.9±5.0%** | 13.1±5.6% |

(c) Expected calibration error (ECE)

| type | ratio | Naive | NaiveB | ZafarCE | ParetoFair CE | ParetoFair BS |
|---|---|---|---|---|---|---|
| sample mean | - | 0.115±0.004 | **0.037±0.003** | 0.219±0.007 | 0.062±0.021 | 0.052±0.012 |
| group mean | 12.5% | 0.288±0.016 | **0.068±0.006** | 0.305±0.013 | 0.094±0.023 | 0.082±0.021 |
| group max | 56.7% | 0.593±0.028 | 0.166±0.047 | 0.561±0.04 | 0.172±0.05 | **0.156±0.05** |
| group min | 0.4% | 0.042±0.002 | **0.025±0.003** | 0.069±0.014 | 0.039±0.013 | 0.032±0.011 |
| disc | 56.3% | 0.552±0.028 | 0.141±0.045 | 0.492±0.045 | 0.133±0.043 | **0.125±0.044** |

Table 1: Performance comparison on MIMIC dataset. We show accuracies and expected calibration error on test set. Standard deviations are computed across 5 splits.

class (deceased, nonwhite, adult). Our methodology produces low discrimination classifiers with high group accuracies, for example, ParetoFair trained with brier score loss (BS) increased the classification accuracy of the most under-served group by over $50\%$ while reducing the accuracy of the best-served group by $12\%$ when compared to the Naive classifier.

### 5.2.2 SKIN LESION CLASSIFICATION

The HAM10000 dataset Tschandl et al. (2018) collects over $10,000$ dermatoscopic images of skin lesions over a diverse population. Lesions are classified according to diagnostic categories. We found that a Naive classifier exhibited almost no measurable discrimination based on age or race on this dataset. We instead chose to use the diagnosis class as both the target and sensitive variable, casting balanced risk minimization as a particular use-case for Pareto fairness. This is possible since our methodology does not require test-time access to sensitive labels. It was not possible to show comparisons against Hardt et al. (2016) since the sensitive attribute is perfectly predictive of the outcome. Table 3 shows accuracies and ECE for all tested methodologies. The Pareto-fair classifier has the overall best calibration results and smallest accuracy disparities across methodologies. Comparisons against other methods were not possible because the target labels are non-binary.

| Groups | ratio | ParetoFair | NaiveB | Naive | ParetoFair | NaiveB | Naive |
|---|---|---|---|---|---|---|---|
| sample mean | - | 78.6% | **84.8%** | 83.6% | **0.055** | 0.065 | 0.095 |
| group mean | 14.3% | 63.4% | **64.1%** | 32.3% | **0.206** | 0.223 | 0.416 |
| group max | 81.0% | 83.5% | 90.9% | **96.0%** | **0.452** | 0.57 | 0.816 |
| group min | 0.5% | **51.9%** | 33.3% | 0.0% | **0.025** | 0.034 | 0.032 |
| disc | 80.4% | **31.7%** | 57.5% | 96.0% | **0.426** | 0.535 | 0.784 |

(a) Accuracy                                      (b) ECE

Table 3: Accuracies (left) and Expected calibration error (right) on test set for HAM10000 dataset.

### 5.2.3 INCOME PREDICTION AND CREDIT RISK ASSESMENT

We tested the proposed method on the Adult UCI dataset Dua & Graff (2017a) and on the German Credit dataset Dua & Graff (2017b). In the Adult UCI dataset the goal is to predict a person's

(a) Adult gender comparison

| type | ratio | Naive | NaiveB | Feldman | Kamishima | Zafar | ParetoFair |
|---|---|---|---|---|---|---|---|
| Male | 67.9% | 92.3±0.4% | 92.3±0.3% | 92.4±0.3% | **92.6±0.4%** | 92.2±0.7% | 92.4±0.3% |
| Female | 32.1% | 80.5±0.4% | 80.3±0.7% | 80.8±0.3% | 80.9±0.4% | 80.8±0.1% | **81.0±0.5%** |
| sample mean | - | 84.3±0.3% | 84.2±0.5% | 84.5±0.3% | 84.6±0.3% | 84.4±0.3% | **84.7±0.3%** |
| group mean | 50.0% | 86.4±0.2% | 86.3±0.4% | 86.6±0.3% | **86.7±0.2%** | 86.5±0.4% | **86.7±0.2%** |
| disc | 35.9% | 11.9±0.7% | 12.0±0.7% | 11.6±0.2% | 11.7±0.7% | **11.4±0.7%** | **11.4±0.6%** |

(b) Adult ethnicity and gender comparison

| type | ratio | Naive | NaiveB | Feldman | Kamishima | Zafar | ParetoFair |
|---|---|---|---|---|---|---|---|
| White male | 39.7% | **90.8±0.3%** | **90.8±0.3%** | 90.6±0.3% | 90.4±1.1% | 90.5±0.6% | 90.7±0.3% |
| Other | 60.3% | 79.8±0.5% | 79.7±0.7% | 80.4±0.6% | 79.4±2.3% | 80.1±0.1% | **80.6±0.6%** |
| sample mean | - | 84.1±0.2% | 84.1±0.4% | 84.5±0.3% | 83.8±1.8% | 84.2±0.2% | **84.6±0.4%** |
| group mean | 50.0% | 85.3±0.1% | 85.2±0.3% | 85.5±0.3% | 84.9±1.6% | 85.3±0.3% | **85.6±0.4%** |
| disc | 20.5% | 11.0±0.8% | 11.1±0.8% | **10.2±0.7%** | 11.0±1.4% | 10.3±0.6% | **10.2±0.6%** |

(c) German comparison

| type | ratio | Naive | NaiveB | Feldman | Kamishima | Zafar | ParetoFair |
|---|---|---|---|---|---|---|---|
| Female | 29.5% | 64.4±6.3% | **64.1±7.5%** | 74.2±7.7% | 68.8±6.8% | 72.7±6.1% | 65.4±5.1% |
| Male | 70.5% | 68.9±5.0% | 70.4±5.3% | 72.0±3.7% | **72.7±2.6%** | 71.6±3.9% | 68.7±6.0% |
| sample mean | - | 67.6±4.5% | 68.6±4.5% | **72.7±4.1%** | 71.6±3.2% | 71.9±3.2% | 67.0±4.4% |
| group mean | 50.0% | 66.6±4.7% | 67.3±4.8% | **73.1±4.9%** | 70.8±4.0% | 72.1±3.5% | 67.5±5.1% |
| disc | 41.0% | 7.6±2.0% | 10.4±3.2% | 6.6±3.3% | 6.0±4.4% | 6.3±4.3% | **5.3±2.8%** |

Table 5: Performance comparison between methods for Adult and German datasets. We show accuracies on test set. Standard deviations are computed across 5 splits.

income, which can be an important factor on meaningful decisions such as credit lending. In the German Credit dataset the goal is predicting credit risk. We select gender and ethnicity as our sensitive attributes. To compare ourselves against state of the art methods Zafar et al. (2017); Feldman et al. (2015); Kamishima et al. (2012) we binarize the sensitive attributes into White-Male and Other when dealing with ethnicity and gender simultaneously, or Male and Female when dealing with gender, and use the unified testbed provided in Friedler et al. (2019). We limit our hypothesis class $\mathcal{H}$ to linear logistic regression to compare evenly against these standard baselines. Results on both datasets are shown in Table 5.

## 6 DISCUSSION

There exists a rich literature of fairness in machine learning in general, and risk-based fairness in particular. Here we explore a relatively untapped sub-problem where the goal is to reduce risk disparity gaps in the most ethical way possible (i.e., minimizing unnecessary harm). Unlike other works in the area, our problem investigates on how to reduce this disparity gap without collecting additional data samples, using the entirety of the available training data to produce an algorithm that is maximally fair with respect to sub-populations, and that does not necessarily require test-time access to sensitive attributes.

We provide a concrete algorithmic adaptation to any standard classification or regression loss to bridge this disparity gap at no unnecessary harm, and demonstrate its performance on several real-world case studies. Even for applications where the need for strict fairness outweighs the need for no-harm classifiers, this methodology can be applied before any post-hoc corrections to ensure that the risk disparity gap is closed in the most risk-efficient way for all involved sub-populations. The proposed algorithm does not sweep through different disparity constraint values, as previously done in related works, making it a simpler alternative.

As an avenue of future research, it could be of interest to analyze if we can automatically identify high-risk sub-populations as part of the learning process and attack risk disparities as they arise, rather than relying on preexisting notions of disadvantaged groups or populations. We strongly believe that no-unnecessary-harm notions of fairness are of great interest for several applications, especially so on domains such as healthcare and lending, where decisions are highly impactful.

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

# A    APPENDIX

## A.1    PROOFS

Here we restate the Lemmas shown in Section 3 along with a sketch of the proofs.

**Lemma 3.1**    If $h \notin \mathcal{P}(\mathcal{H}, \mathcal{A}) \to \exists h_p \succ h \in \mathcal{P}(\mathcal{H}, \mathcal{A}) : R_a(h_p^{ER}) \leq R_a(h^{ER}), \forall a$, where $h^{ER}$ is a equality of risks classifier in $\mathcal{H}$ such that $R_a(h^{ER}) = \max_{a' \in \mathcal{A}} R_{a'}(h), \forall a$ and $h_p^{ER} : R_a(h_p^{ER}) = \max_{a' \in \mathcal{A}} R_{a'}(h_p), \forall a$.

*Proof.*  If $h_p$ dominates $h \to R_a(h_p^{ER}) = \max_{a' \in \mathcal{A}} R_{a'}(h_p) \leq \max_{a' \in \mathcal{A}} R_{a'}(h) = R_a(h^{ER}) \,\forall a \in \mathcal{A}$.    □

**Definition A.1.**  Dominant vector: A vector $r' \in \mathbb{R}^k$ is said to dominate $r \in \mathbb{R}^k$, noted as $r' \succ r$, if $r_i \geq r_i', \forall i = 1, ..., k$ and $\exists j : r_j > r_j'$ (i.e., strict inequality on at least one component).

**Definition A.2.**  Pareto field: A function $P : \Omega \to \mathbb{R}$ is defined as a *Pareto field* over a convex set $\Omega \subset \mathbb{R}^k$ if $P \in \mathcal{C}^1$ is a continuously differentiable function such that $\nabla_i P(r) > 0 \,\forall r \in \Omega, \forall i = 1, \dots, k$.

**Lemma A.1.**  *Let $\Omega \subset \mathbb{R}^k$ be a convex set, and $P : \Omega \to \mathbb{R}$ a Pareto field. Then the set $D = \{r \in \Omega : P(r) = 0\}$ is a proper Pareto set in $\Omega$, and the set $D^+ = \{r \in \Omega : P(r) > 0\}$ is the set of dominated points, i.e., $D^+ = \{r \in \Omega : \exists r' \in D \mid r \succ r'\}$.*

*Proof.*  First we prove that $D = \{r \in \Omega : P(r) = 0\}$ is a Pareto set. Assume by contradiction that there exists $r, r' \in D \mid r' \succ r$, then $r' - r = c = \sum_i \delta_i e_i \quad \delta_i \geq 0 \forall i \in \{1, \dots, k\}, \exists j \in \{1, \dots, k\} \mid \delta_j > 0$. $e_i$ is a standard basis vector.

Using the Gradient theorem we have

$$
\begin{aligned}
P(r') &= P(r) + \int_{r \to r'} \nabla P(\lambda) d\lambda, \\
&= P(r) + \int_0^1 \langle (\delta_1, \dots, \delta_k), (\nabla P_0(r + \lambda c), \dots, \nabla P_k(r + \lambda c)) \rangle d\lambda, \\
&= P(r) + \sum_i \delta_i \int_0^1 \underbrace{\nabla P_i(r + \lambda c)}_{>0} d\lambda, \\
&> P(r),
\end{aligned}
$$

which directly contradicts $r', r \in D$. No point in set $D$ is dominated by another point in set $D$, making set $D$ a proper Pareto set.

To show that the set $D^+$ is the set of dominated points, we first note that for all $r \in \Omega \mid r' \in D, r \succ r'$ we have $P(r) > P(r')$ using the same arguments as above, meaning that the set of dominated points is included in $D^+$. Similarly, for all $r \in \Omega \mid r' \in D, r' \succeq r$ we have $P(r') \geq P(r)$, meaning that non-dominated points are not a part of set $D^+$.    □

**Lemma A.2.**  *Let $\Omega \subset \mathbb{R}^{|\mathcal{A}|}$ be a convex set defined by $\Omega = \{r \in \mathbb{R}^{|\mathcal{A}|} : g^c(r) \geq 0 \forall c \in \{1, \dots, C\}, g^c$ continuously differentiable$\}$; let $P : \Omega \to \mathbb{R}$ be a convex Pareto field with corresponding Pareto set $D = \{r \in \Omega : P(r) = 0\}$. Let $r^* \in D$ and $\phi(r, \mu) = \sum_{i=1}^{|\mathcal{A}|} r_i + \mu_i(r_i - c)^{2+}$, with $c < r^*$. There exists a set of $\mu^* \succ 0$ such that:*

$$
r^* = \arg\min_{r \in \mathbb{R}^{|\mathcal{A}|}} \phi(r, \mu^*) \quad s.t. : P(r) \geq 0, g^c(r) \geq 0 \,\forall c \in \{1, \dots, C\}
$$

.

*Proof.*  By hypothesis, both $\Omega$ and $\{r \in \Omega : P(r) \geq 0\}$ are convex, so the intersection is also convex. For $\mu \succ 0$, $\phi(r, \mu)$ is a convex function of $r$. Under these hypothesis, the Karush-Kuhn-Tucker (KKT) conditions are necessary and sufficient to recover a global minimizer. So it suffices to find a set $\mu^*$ such that the KKT conditions are exactly satisfied at $r^*$.

Let $J^* = \{j : g^j(\boldsymbol{r}^*) = 0\}$ be the set of indices of active $g^c$ constraints at $\boldsymbol{r}^*$, note that by hypothesis, $\boldsymbol{r}^* \in D$ and therefore $P(\boldsymbol{r}^*)$ is always active. Focusing on the first KKT condition on the active set we get $\nabla \phi(\boldsymbol{r}^*, \vec{\mu}) = \lambda \nabla P(\boldsymbol{r}^*) + \sum_{j \in J^*} \eta_j \nabla g^j(\boldsymbol{r}^*)$, in matrix form:

$$
\begin{bmatrix} 1 \\ \vdots \\ 1 \end{bmatrix} + \begin{bmatrix} 2(r_1^* - c)^+ & & 0 \\ & \ddots & \\ 0 & & 2(r_A^* - c)^+ \end{bmatrix} \mu = \begin{bmatrix} | & | & & | \\ \nabla P & \nabla g^{j_1} & \dots & \nabla g^{j_{|J^*|}} \\ | & | & & | \end{bmatrix} \begin{bmatrix} \lambda \\ \eta_1 \\ \vdots \\ \eta_{c^*} \end{bmatrix}.
$$

By hypothesis, we have $r_i - c > 0$ and therefore

$$
\mu_i = \frac{\lambda \nabla P_i(r^*) + \sum_{j \in J^*} \eta_j \nabla g_i^j(r^*) - 1}{2(r_i^* - c)}.
$$

In particular, for $\eta_j = 0 \; \forall j$ and $\mu_1 > [\frac{\nabla P_i(r^*)}{\nabla P_1(r^*)} - 1] \frac{1}{2(r_1^* - c)^+} \forall i \neq 1$ we get

$$
\mu_i = [(\mu_1 2(r_1^* - c)^+ + 1) \frac{\nabla P_i(r^*)}{\nabla P_1(r^*)} - 1] \frac{1}{2(r_i^* - c)^+} > 0 \forall i \neq j,
$$

$$
\mu_1 > 0,
$$

$$
\lambda = \frac{\mu_1 2(r_1^* - c)^+ + 1}{\nabla P_1(r^*)} > 0,
$$

which satisfies the KKT conditions and produces the required minimizer $\boldsymbol{r}^*$ for all points $\boldsymbol{r}^* \in D$.

$\square$

**Lemma A.3.** *Let $\Omega \subset \mathbb{R}^2$ be a convex set defined by $\Omega = \{\boldsymbol{r} \in \mathbb{R}^2 : g^c(\boldsymbol{r}) \geq 0 \forall c \in \{1, \dots, C\}, g^c$ continuously differentiable\}; let $P : \Omega \to \mathbb{R}$ be a convex Pareto field with corresponding Pareto set $D = \{\boldsymbol{r} \in \Omega : P(\boldsymbol{r}) = 0\}$. Let $\epsilon > 0$, $\boldsymbol{r}^* = \underset{\boldsymbol{r} \in D}{\arg\min}[\max_i r_i - \min_j r_j]$*

*then*

$$
c = \min(r_1, r_2) - \epsilon < \min(r_1^*, r_2^*) \; \forall (r_1, r_2) \in D.
$$

*Proof.* **case $r_1^* = r_2^*$:** Assume $\boldsymbol{r} \in D : \min_i r_i > r_1^*$. Then $\boldsymbol{r} \succ \boldsymbol{r}^*$, which contradicts the hypothesis $\boldsymbol{r} \in D$.

**case $r_1^* \neq r_2^*$:** Without loss of generality, we can take $r_1^* > r_2^*$. We can then prove by couterpositive that $\boldsymbol{r} \in D$ implies $r_1 > r_2$. Assume by contrapositive that there exists $\boldsymbol{r}' \in D : r_2' > r_1'$. By hypothesis $D$ is a continuous curve, and we can find a path $R(t) : R(0) = \boldsymbol{r}*, R(1) = \boldsymbol{r}', R(t) \in D \forall t \in [0, 1]$, which would imply there exists $\hat{t} : R(\hat{t}) = (\hat{r}, \hat{r})$, in which case $r_1^* = r_2^*$, which contradicts $r_1^* \neq r_2^*$.

We then have that for $\boldsymbol{r} \in D$, $r_1 > r_2$. Assume by counterpositive that $r_2 = min(r_1, r_2) > min(r_1^*, r_2^*) = r_2^*$ then $r_1 < r_1^*$ (since both $\boldsymbol{r}, \boldsymbol{r}^* \in D$), but then $|r_1 - r_2| < |r_1^* - r_2^*|$ which contradicts the hypothesis.

Putting both statements together, we proved that if $r_1^* > r_2^*$, then $\forall \boldsymbol{r} \in D, \min(r_1, r_2) = r_2 \leq min(r_1^*, r_2^*) = r_2^*$.

$\square$

**Lemma A.4.** *Let $\Omega \subset \mathbb{R}^2$ be a convex set defined by $\Omega = \{\boldsymbol{r} \in \mathbb{R}^2 : g^c(\boldsymbol{r}) \geq 0 \forall c \in \{1, \dots, C\}, g^c$ continuously differentiable\}; let $P : \Omega \to \mathbb{R}$ be a convex Pareto field with corresponding Pareto set $D = \{\boldsymbol{r} \in \Omega : P(\boldsymbol{r}) = 0\}$. Let $\mu = (\mu_1, \mu_2) > 0$, $\mu' = (\mu_1, \mu_2')$; $\mu_2' > \mu_2$. Define $\phi(\boldsymbol{r}, \mu, c) = \sum_{i=1}^2 r_i + \mu_i(r_i - c)^{2+}$ and*

$$
\boldsymbol{r} = \underset{\substack{s.t. \, P(r) \geq 0 \\ \boldsymbol{r} \in \Omega}}{\arg\min} \phi(\boldsymbol{r}, \mu, c),
$$

$$
\boldsymbol{r}' = \underset{\substack{s.t. \, P(\boldsymbol{r}) \geq 0 \\ \boldsymbol{r} \in \Omega}}{\arg\min} \phi(\boldsymbol{r}, \mu', c).
$$

*With $c < r_2$, then, $r_1' \geq r_1$, $r_2' \leq r_2$, with equality if and only if $\boldsymbol{r}' = \hat{r} = \underset{s.t. \, \boldsymbol{r} \in \{P(\boldsymbol{r}) \geq 0\} \cap \Omega}{\arg\min} r_2$*

*Proof.* By hypothesis, we have

$$\phi(\boldsymbol{r}, \mu', c) \geq \phi(\boldsymbol{r}', \mu', c)$$

$$\phi(\boldsymbol{r}, \mu, c) + (\mu_2' - \mu_2)(r_2 - c)^{2+} \geq \phi(\boldsymbol{r}', \mu, c) + (\mu_2' - \mu_2)(r_2' - c)^{2+}$$

$$\phi(\boldsymbol{r}, \mu, c) \geq \phi(\boldsymbol{r}', \mu, c) + (\mu_2' - \mu_2)[(r_2' - c)^{2+} - (r_2 - c)^{2+}]$$

since by hypothesis $\phi(\boldsymbol{r}, \mu, c) \leq \phi(\boldsymbol{r}', \mu, c)$, it follows $r_2' \leq r_2$, and therefore $r_1' \geq r_1$, because they both belong to the Pareto set.

**Case $\boldsymbol{r}' = \boldsymbol{r}$.** To analyze when the equality arises, note that the tangent to $D$ at $\boldsymbol{r}$ is $\tau(\boldsymbol{r}) = (\nabla_2 P(\boldsymbol{r}), -\nabla_1 P(\boldsymbol{r}))$ and that $\boldsymbol{r} + \delta\tau(\boldsymbol{r})$, $\delta > 0$ is a valid search direction if $\boldsymbol{r} \neq \hat{\boldsymbol{r}}$ ($\boldsymbol{r}$ does not minimize $r_2$). Also note that $\nabla\phi(\boldsymbol{r}, \mu', c) = \nabla\phi(\boldsymbol{r}, \mu, c) + (0, 2(\mu_2' - \mu_2)(r_2 - c)^+)$

By contradiction, assume $\boldsymbol{r} \neq \hat{\boldsymbol{r}}$, then $\boldsymbol{r} + \delta\tau(\boldsymbol{r})$, $\delta > 0$ is a valid search direction and $\langle(\delta\tau(\boldsymbol{r})), \nabla\phi(\boldsymbol{r}, \mu', c\rangle > 0$, which implies $\boldsymbol{r}' \neq \underset{\substack{s.t.\ P(r)\geq 0 \\ r\in\Omega}}{\arg\min} \phi(r, \mu', c)$ and contradicts the hypothesis.

Therefore, if $\boldsymbol{r}' = \boldsymbol{r}$ it follows $\boldsymbol{r}' = \boldsymbol{r} = \hat{\boldsymbol{r}}$

$\square$

**Lemma A.5.** *Let $\Omega \subset \mathbb{R}^{|\mathcal{A}|}$ be a convex set defined by $\Omega = \{r \in \mathbb{R}^{|\mathcal{A}|} : g^c(r) \geq 0 \,\forall c \in \{1, \ldots, C\}, g^c$ continuously differentiable\}; let $P : \Omega \to \mathbb{R}$ be a convex Pareto field with corresponding Pareto set $D = \{r \in \Omega : P(r) = 0\}$. Let $r^* = \underset{r\in D}{\arg\min}(\max_i r_i - \min_j r_j)$ and $\phi(r, \mu, c) = \sum_{i=1}^{|\mathcal{A}|} r_i + \mu_i(r_i - c)^{2+}$. Let $\mu^0 > 0$ be an initial penalty vector, $r^0 = \underset{r\in D}{\arg\min}\sum_i r_i$, $\epsilon > 0$, $c^0 = \min_i r_i^0 - \epsilon, \gamma > 0$, $\xi \in (0, 1)$. Define the following auxiliary variables*

$$c^{k+1} = \min_i r_i^k - \epsilon$$

$$\Gamma^k = \max_i r_i^k - \min_i r_i^k$$

$$a^\dagger = \underset{a}{\arg\max}(r_a^k)$$

$$\mu_{a^\dagger}^\dagger = (\gamma + 1)\mu_{a^\dagger}^k$$

$$\mu_{\backslash a^\dagger}^\dagger = \mu_{\backslash a^\dagger}^k \tag{2}$$

$$r^\dagger = \underset{\substack{s.t.\ P(r)\geq 0 \\ r\in\Omega}}{\arg\min}\phi(r, \mu^\dagger, c^{k+1})$$

$$c^\dagger = \min_i r_i^\dagger - \epsilon$$

$$\Gamma^\dagger = \max_i r_i^\dagger - \min_j r_j^\dagger$$

*We define the following iteration procedure*

compute Eq 2

If $\Gamma^k = \Gamma^\dagger$

terminate

While $\Gamma^\dagger > \Gamma^k$

$$\gamma \leftarrow \xi\gamma$$

Recompute Eq 2

End While

$$\mu^{k+1} = \mu^\dagger \cdot \frac{(r^\dagger - c^k)^+}{(r^\dagger - c^\dagger)^+}$$

$$r^{k+1} = r^\dagger$$

*The procedure is convergent to $r^*$ for $|\mathcal{A}| = 2$.*

*Proof.* From Lemma A.3 we have $c_k < \min_i r_i^* \ \forall k$, we can then apply Lemma A.2 and state that there exists a set $\mu^{*k}$ such that $r^* = \underset{\substack{\text{s.t. } P(r) \geq 0 \\ r \in \Omega}}{\arg\min} \ \phi(r, \mu^{*k}, c^k)$.

We split the proof in two scenarios, one where $\Gamma^* = 0$ and then $\Gamma^* > 0$, since they show different convergence properties.

**case $\Gamma^* = 0$.** Assume without loss of generality that $r_1^k = \arg\max_i r_i^k$, from Lemma A.4 we have $\mu_1^k < \mu_1^{k*}$, and further, for all $\mu_1^\dagger \in (\mu_1^k, \mu_1^{k*})$, we have $r_1^k < r_1^\dagger < r_1^*$. Since $|\mathcal{A}| = 2$ and $r^k, r^\dagger, r^*$ all belong to $D$, we also have $r_2^k > r_2^\dagger > r_2^*$, which implies $\Gamma^k > \Gamma^\dagger > \Gamma^* = 0$. The subiteration procedure is always successful, and we get a sequence of iterates such that $\Gamma^k > \Gamma^{k+1}$, a strictly decreasing sequence bounded below by 0, which implies asymptotic convergence to $r^*$. Procedure never terminates with probability 1 (event $\mu_k \in \mu_k^*$ has probability 0).

**case $\Gamma^* > 0$** Assume without loss of generality that $r_1^* < r_2^*$. Following the arguments from Lemma A.3 we observe that $r_1 < r_2 \ \forall (r_1, r_2) \in D$. In these conditions, we have that $\frac{\mu_2^k}{\mu_1^k} < \frac{\mu_2^{k+1}}{\mu_1^{k+1}}$ and $c^{k+1} > c^k$. We will re-derive the KKT conditions in this scenario to show that convergence is guaranteed for $\mu_2^k > \mu_{2*}$, and that this $\mu_{2*}$ is independent of $c^k$, meaning that convergence to the optimal $r^*$ can be achieved in a finite number of steps.

Let $\nabla P(r^*) = (dP_1, dP_2) > 0$, with corresponding tangent vector $\tau^* = (dP_2, -dP_1)$, for $r^*$ to be the risk with minimal gap, $r^* + \delta\tau^*$ must be an infeasible descent direction for $\delta > 0$. Therefore, there is an active constraint from $\Omega$, $g(r)$, with $\nabla g(r^*) = (dg_1, dg_2)$ such that $\langle (dg_1, dg_2), (dP_2, -dP_1) \rangle < 0$.

Deriving the first KKT conditions, we get

$$1 + \mu_1 2(r_1^* - c) = \lambda dP_1 + \eta dg_1,$$
$$1 + \mu_2 2(r_2^* - c) = \lambda dP_2 + \eta dg_2,$$

and in terms of $\lambda, \eta$

$$\lambda = \frac{1 + \mu_1 2(r_1^* - c) - \eta dg_1}{dP_1},$$
$$\eta = \frac{dP_1(1 + \mu_2 2(r_2^* - c) - dP_2(1 + \mu_1 2(r_1^* - c))}{dg_2 dP_1 - dP_2 dg_1}.$$

Note that $dg_2 dP_1 - dP_2 dg_1 > 0$ from $\langle (dg_1, dg_2), (dP_2, -dP_1) \rangle < 0$. To recover a valid set of Lagrange multipliers $\eta, \lambda > 0$, it suffices to have

$$dP_1(1 + \mu_2 2(r_2^* - c)) > dP_2(1 + \mu_1 2(r_1^* - c)),$$
$$\mu_2 > \frac{(1 + \mu_1 2(r_1^* - c))\frac{dP_2}{dP_1} - 1}{2(r_2^* - c)},$$
$$\mu_2 > \frac{(1 + \mu_1 2(r_1^* - c^0))\frac{dP_2}{dP_1} - 1}{2\epsilon},$$

where we used that $c^0 < c^k < \epsilon \ \forall k$. Therefore, for $\frac{\mu_2^k}{\mu_1^k}$ large enough, $r^k = r^*$ and the algorithm terminates. $\square$

## A.2 Algorithmic details

Here we provide details on how we optimize our adaptive loss in-between penalty update ($\mu_a$) steps, shown in Algorithm 2.

## A.3 Methods

We compare the performance of the following methods:

---

**Algorithm 2:** AdaptiveOptimize

---

**Given:** $h_\theta, L, \vec{\mu}, c, \mathcal{D}^{\text{Tr}}, \mathcal{D}^{\text{Val}}, n_p, \text{lr}, \text{B}$
$p \leftarrow 0, \quad \phi^* \leftarrow \infty \quad h^* \leftarrow h_\theta$
**while** $p \leq n_p$                                             // Making loss progress
**do**
$\quad$ **for** $\{(x_i, y_i, a_i)\}_{i=1}^B \in \mathcal{D}^{\text{Tr}}$          // Run one epoch of SGD on training set
$\quad$ **do**
$\quad\quad I_a = \{i \in \{1, \ldots, B\} \wedge a_i = a\}, \quad \bar{R}_a^{\text{B}}(h_\theta) = \frac{1}{|I_a|} \sum\limits_{i \in I_a} L(h_\theta(x_i), y_i)$  // Empirical risks
$\quad\quad \phi^B(h_\theta) = \sum_{a \in \mathcal{A}} \bar{R}_a^{\text{B}}(h_\theta) + \mu_a(\bar{R}_a^{\text{B}}(h_\theta) - c)^{+2}$
$\quad\quad \theta \leftarrow \theta - lr \nabla_\theta \phi^B(h_\theta)$                                    // Gradient step
$\quad$ **end**
$\quad$ **if** $\phi^{\text{Val}}(h_\theta) < \phi^*$        // Evaluate improvement on Val and update target risks
$\quad$ **then**
$\quad\quad h^* \leftarrow h_\theta; \quad \phi^* \leftarrow \phi^{\text{Val}}(h_\theta); \quad p \leftarrow 0$
$\quad$ **else**
$\quad\quad p \leftarrow p + 1;$
$\quad$ **end**
$\quad R^{\text{Val}} \leftarrow \bar{R}^{\text{Val}}(h)$
**end**
**Return:** $h^*, R^{\text{Val}}$

---

**Kamishima.**   Kamishima et al. (2012) uses logistic regression as a baseline classifier, and requires numerical input (observations), and binary sensitive attribute and target variable. Fairness is controlled via a regularization term with a tuning parameter $\eta$ that controls trade-off between fairness and overall accuracy. $\eta$ is optimized via grid search with $\eta \in (0, 300)$ as in the original paper. We report results on the hyperparameter configuration that produces the smallest accuracy disparity between sensitive subgroups.

**Feldman.**   Feldman et al. (2015) provides a preprocessing algorithm to sanitize input observations. It modifies each input attribute so that the marginal distribution of each coordinate is independent on the sensitive attribute. The degree to which these marginal distributions match is controlled by a $\lambda$ parameter between 0 and 1. It can handle numerical and categorical observations, as well as non-binary sensitive attributes, and arbitrary target variables. Following Friedler et al. (2019), we train a linear logistic regressor on top of the sanitized attributes. $\lambda$ is optimized via grid search with increments of 0.05. We report results on the hyperparameter configuration that produces the smallest accuracy disparity between sensitive subgroups.

**Zafar.**   Zafar et al. (2017) Addresses disparate mistreatment via a convex relaxation. Specifically, in the implementation provided in Friedler et al. (2019), they train a logistic regression classifier with a fairness constraint that minimizes the covariance between the sensitive attribute and the classifier decision boundary. This algorithm can handle categorical sensitive attributes and binary target variables, and numerical observations. The maximum admissible covariance is handled by a hyperparameter $c$, tuned by logarithmic gridsearch with values between 0.001 and 1. We report results on the hyperparameter configuration that produces the smallest accuracy disparity between sensitive subgroups.

**Hardt.**   Hardt et al. (2016) proposes a post-processing algorithm that takes in an arbitrary predictor and the sensitive attribute as input, and produces a new, fair predictor that satisfies equalized odds. This algorithm can handle binary target variables, an arbitrary number of sensitive attributes, and any baseline predictor, but requires test-time access to sensitive attributes. it does not contain any tuning parameter. We apply this method on top of both the Naive Classifier and our Pareto Fair classifier.

**Naive Classifier (Naive).**   Standard classifier, trained to minimize an expected risk $h = \arg\min_{h \in \mathcal{H}} E_{X,A,Y}[L(h(X), Y)]$. The baseline classifier class $\mathcal{H}$ is implemented as a neural network and varies by experiment as described in Section A.6, the loss function also varies by experiment and is also described in Section A.6. Optimization is done via stochastic gradient descent.

**Naive Balanced (NaiveB).** Baseline classifier designed to address undersampling of minority classes, trained to mimimize a class-rebalanced expected risk $h = \arg\min_{h \in \mathcal{H}} E_{A \sim U[1,...,|\mathcal{A}|],(X,Y) \sim P(X,Y|A)}[L(h(X),Y)]$. Like the Naive classifier, it is implemented as a neural network and optimized via stochastic gradient descent. The sole difference with the Naive classifier is that, during training, training samples are drawn from the new input distribution $A \sim U[1,...,|\mathcal{A}|]$; $X,Y|A \sim P(X,Y|A)$, which is achieved by re-weighted sampling of the original training dataset.

**Pareto Fair.** Our proposed methodology, trained to minimize an adaptive loss function using the procedure described in Algorithm 1. Addresses risk disparity minimization without introducing unnecessary harm to any subgroup. The baseline classifier class $\mathcal{H}$ is implemented as a neural network and varies by experiment as described in Section A.6, the loss function also varies by experiment and is also described in Section A.6.

## A.4 EVALUATION METRICS

Here we describe the metrics used to evaluate the performance of all tested methods. We are given a set of test samples $\mathcal{D}_t = \{(x_i, y_i)\}_{i=1}^N$ where $x_i \in \mathcal{X}$ is a realization of our model input and $y_i \in \mathcal{Y}$ the corresponding objective. We assume that $\mathcal{Y}$ is a finite alphabet, as in a classification problem, and we will represent the one-hot encoding of $y_i$ as $\vec{e}_i$. Given a trained model $h : \mathcal{X} \to [0,1]^{|\mathcal{Y}|}$ the predicted output for a an input $x_i$ is a vector $h(x_i) = \vec{p}_i$ such that $(\vec{p}_i)_j \in [0,1], \forall j \in \{1,...,|\mathcal{Y}|\}$ and $|\sum_{j=1}^{|\mathcal{Y}|}(\vec{p}_i)_j = 1$ (e.g.: output of a softmax layer). The predicted class is $\hat{y}_i = \arg\max_j(\vec{p}_i)_j$ and its associated confidence is $\hat{p}_i = \max_j(\vec{p}_i)_j$. Ideally $\hat{y}_i$ should be the same as $y_i$. Using these definitions, we compute the following metrics.

**Accuracy (AC):** $\frac{1}{N}\sum_{i=1}^N \mathbf{1}(y_i = \hat{y}_i)$. Fraction of correct classifications in dataset.

**Confidence (CO):** $\frac{1}{N}\sum_{i=1}^N \hat{p}_i$. Average magnitude of the predicted class probability.

**Brier Score (BS):** $\frac{1}{N}\sum_{i=1}^N ||\vec{e}_i - \vec{p}_i||^2$ where $\vec{e}_i$ is the one-hot representation of the categorical ground truth value $y_i$. This quantity is also known as Mean square error (MSE).

**Cross-Entropy (CE):** $-\frac{1}{N}\sum_{i=1}^N \sum_{j=1}^{|\mathcal{Y}|}(\vec{e}_i)_j log(\vec{p}_i)_j$ also known as negative log-likelihood (NLL) of the multinomial distribution.

**Expected Calibration Error (ECE):** $\frac{1}{N}\sum_{m=1}^M \big|\sum_{i \in B_m}[\mathbf{1}(y_i = \hat{y}_i) - \hat{p}_i]\big|$ where $M$ is the number of bins to divide the interval $[0,1]$ such that $B_m = \{i \in \{1,..,N\} : \hat{p}_i \in (\frac{m-1}{M}, \frac{m}{M}]\}$ are the group of samples that our model assigns a confidence $(\hat{p}_i)$ in the interval $(\frac{m-1}{M}, \frac{m}{M}]$. Measures how closely the predicted probabilities match the true base rates.

**Maximum Calibration Error (MCE):** $max_{m \in \{1,...,M\}}\big|\frac{1}{|B_m|}\sum_{i \in B_m}[\mathbf{1}(y_i = \hat{y}_i) - \hat{p}_i]\big|$. Measures worst-case miscalibration errors.

These metrics are computed independently for each sensitive subgroup on the test set and reported in Section A.7.

## A.5 EXPERIMENTS ON REAL DATA

The following is a description of the data and experiments for each of the real datasets. The information present here is summarized in Table 7.

| Dataset | Outcome/ Objective | Sensitive Attribute | Train/Val/Test | splits |
|---------|-------------------|---------------------|----------------|--------|
| Adult Dua & Graff (2017a) | Income 2 categories | Gender (F/M) Ethnicity(W/NW) 2x2 categories | 60/20/20 | 5 |
| German Dua & Graff (2017a) | Credit 2 categories | Gender (F/M) 2 categories | 60/20/20 | 5 |
| MIMIC-III Johnson et al. (2016) | Mortality (A/D) 2 categories | Mortality(A/D), Age (A/S) Ethnicity (W/NW) 2x2x2 categories | 60/20/20 | 5 |
| HAM10000 Tschandl et al. (2018) | Type of lesion 7 categories | Type of lesion 7 categories | 80/10/10 | 1 |

Table 7: Basic characteristics of real datasets

**MIMIC-III.** This dataset consist of clinical records collected from adult ICU patients at the Beth Israel Deaconess Medical Center (MIMIC-III dataset) Johnson et al. (2016). The goal is predicting patient mortality from clinical notes. We follow the pre-processing methodology outlined in Chen et al. (2018), where we analyze clinical notes acquired during the first 48 hours of ICU admission; discharge notes where excluded, as where ICU stays under 48 hours. Tf-idf statistics on the $10,000$ most frequent words in clinical notes are taken as input features.

We identify 8 sensitive groups as the combination of age (under/over 55 years old), ethnicity as determined by the majority group (white/nonwhite) and outcome (alive/deceased). Here we will use the term adult to refer to people under 55 years old and senior otherwise. This dataset shows large sample disparities since 56.7% of corresponds to the overall majority group (alive-senior-white) and only 0.4% to the overall minority group (deceased-adult-nonwhite).

We used a fully connected neural network as described in table 8 as the baseline classifier for our proposed Pareto Fair algorithm. We compare our results against both the Naive and Naive Balanced algorithms using the same neural network architecture, and crossentropy (CE) as our training loss. We also evaluate the performance of Zafar applied on the feature embeddings learned by the Naive Balanced classifier (Results for Zafar over the original input features were not promising).

We report the performance across a 5-fold split of the data, we used a 60/20/20 train-validation-test partition as described on Table 7 and report results over the test set. We denote the overall sensitive attribute as the combination of outcome (A:alive/D:deceased), age (A:adult/S: senior) and ethnicity (W:white, NW:nonwhite) with shorthand notation of the form D/A/W to denote, for example deceased, white adult. We also note that results on both Zafar and Hardt were done over only the sensitive attributes Adult/Senior and White/Nonwhite, outcome was not considered as a sensitive attribute for both methods since Hardt requires test-time access to sensitive attributes, which would not be possible in this case, and Zafar attempts to decorrelate sensitive attributes and classification decision boundaries, which is counterproductive when the sensitive attribute includes the correct decision outcome.

**HAM10000.** This dataset contains over $10,000$ dermatoscopic images of skin lesions over a diverse population Tschandl et al. (2018). Lesions are classified in 7 diagnostic categories, and the goal is to learn a model capable of identifying the category from the lesion image. The dataset is highly unbalanced since 81% of the samples correspond to a melanocytic nevi lesion (nv), and 0.5% to dermatofibroma (df).

Here we chose to use the diagnosis class as both the target and sensitive variable, casting balanced risk minimization as a particular use-case for Pareto fairness.

We load a pre-trained DenseNet121 network Huang et al. (2017) and train it to classify skin lesions from dermatoscopic images using our Pareto fairness algorithm. We compared against the Naive and the Naive balanced training setup. Note that in the naive balance approach we use a batch sampler where images from each class have the same probability, this can be seen as a naive oversampling technique. Table 8 shows the details of the implementation.

We used the original train-validation-test (80/10/10) split, and report results on the test set. Notation for each group follows the original notation: Actinic keratoses and intraepithelial carcinoma / Bowen's disease (akiec), basal cell carcinoma (bcc), benign keratosis-like lesions (bkl), dermatofibroma (df), melanoma (mel), melanocytic nevi (nv) and vascular lesions (vasc).

**Adult.**   The Adult UCI dataset Dua & Graff (2017a) is based on the 1994 U.S. Census and contains data on $32, 561$ adults. The data contains 105 binarized observations representing education status, age, ethnicity, gender, and marital status and a target variable indicating income status (binary attribute representing over or under $\$50, 000$). Following Friedler et al. (2019), we take ethnicity and gender as our target sensitive attributes, defining two subgroups (White Males and Other). We also present results taking just the gender as sensitive attribute (Male/Female). To compare our Pareto Fair algorithm evenly against the other methods, we limit our hypothesis class to linear logistic regression.

**German.**   The German credit dataset Dua & Graff (2017a) contains 20 observations collected across 1000 individuals, and a binary target variable assessing the individual's credit score as good or bad. We take gender (Male/Female) as the sensitive attribute, which is not included in the data but can be inferred. As in the Adult dataset, we limit our hypothesis class to linear logistic regression to compare evenly across methodologies.

## A.6   NEURAL ARCHITECTURES AND PARAMETERS

Table 8 summarizes network architectures and loss functions for all experiments in Section 5. Note that all networks have a standard dense softmax as their final layer. The training optimizer is standard ADAM Kingma & Ba (2014), loss functions were either crossentropy (CE) or brier score (BS), also known as categorical mean square error (MSE).

| Dataset | Network Body | Gate | Loss type | Parameters |
|---|---|---|---|---|
| Synthetic | FullyConnected 64x64 | ELU | BS | batch size=32 lr=5e-4 |
| Adult - German Dua & Graff (2017a) | Logistic Regression (LR) | - | CE | batch size=32 lr=5e-4 |
| MIMIC-III Johnson et al. (2016) | FullyConnected 2048x2048 | ELU | CE/BS | batch size=512 lr=1e-6/5e-6 |
| HAM10000 Tschandl et al. (2018) | DenseNet121 Huang et al. (2017) | ReLU | BS | batch size=32 lr=5e-6 |

Table 8: Summary of network architectures and losses. All networks have a final softmax dense layer as the output layer.. CE refers to crossentropy and BS to Brier score or categorical MSE. Training was done using ADAM optimizer with the learning rates (lr) specified in the table.

## A.7   SUPPLEMENTARY RESULTS

The following tables show a quantitative performance comparisons between the various fairness methods described in Section A.3 over the datasets described in Section A.5. All metrics presented are described in Section A.4 and computed per sensitive attribute. We additionally provide results for the overall mean across the dataset (sample mean), as well as results on the group-balanced mean across groups (group mean). We use group max and group min to denote the maximum and minimum metric values attained over all sensitive groups. We denote group max - group min as Discrimination, which is the worst case difference in performance across groups. When the dataset contains multiple splits, we report both split mean and split standard deviation.

| type | ratio | Naive | NaiveB | ZafarCE | ParetoFair CE | ParetoFair BS |
|---|---|---|---|---|---|---|
| A/A/NW | 5.7% | 99.1±0.4% | 86.3±1.5% | 93.0±1.4% | 80.4±2.8% | 83.4±2.6% |
| A/A/W | 13.3% | 98.8±0.5% | 86.3±1.1% | 90.0±1.3% | 81.0±1.5% | 83.2±1.5% |
| A/S/NW | 12.9% | 97.5±0.6% | 76.5±1.7% | 81.8±1.7% | 69.5±2.2% | 71.4±3.0% |
| A/S/W | 56.7% | 97.9±0.3% | 79.0±0.6% | 77.4±0.7% | 73.5±1.3% | 74.6±1.6% |
| D/A/NW | 0.4% | 23.4±9.4% | 76.1±8.5% | 47.7±9.6% | 76.6±5.3% | 78.6±6.1% |
| D/A/W | 0.9% | 32.6±3.7% | 80.1±3.3% | 60.5±6.3% | 85.0±3.8% | 83.3±4.2% |
| D/S/NW | 1.8% | 21.4±2.2% | 66.9±2.4% | 48.2±2.0% | 72.6±5.1% | 73.3±2.9% |
| D/S/W | 8.3% | 23.4±2.2% | 67.4±1.9% | 57.1±2.2% | 71.0±4.2% | 72.5±3.6% |
| sample mean | - | **89.5±0.2%** | 78.9±0.7% | 78.0±0.7% | 74.3±0.8% | 75.7±1.1% |
| group mean | 12.5% | 61.8±1.5% | 77.3±1.3% | 69.4±1.3% | 76.2±0.5% | **77.5±0.7%** |
| group max | 56.7% | **99.2±0.3%** | 88.1±1.3% | 93.0±1.4% | 86.4±2.2% | 87.1±2.0% |
| group min | 0.4% | 18.1±2.5% | 65.7±1.2% | 43.7±3.9% | 67.9±2.5% | **69.9±2.2%** |
| disc | 56.3% | 81.1±2.5% | 22.5±2.5% | 49.3±4.4% | 18.5±4.2% | **17.1±2.6%** |

| type | ratio | NaiveB H | ZafarCE H | ParetoFair CE H | ParetoFair BS H |
|---|---|---|---|---|---|
| A/A/NW | 5.7% | 76.3±1.9% | 68.4±2.2% | 69.9±2.5% | 71.6±3.2% |
| A/A/W | 13.3% | 76.7±1.6% | 67.8±1.3% | 70.8±2.4% | 71.8±1.9% |
| A/S/NW | 12.9% | 76.4±2.2% | 67.9±2.1% | 70.1±2.5% | 71.3±3.1% |
| A/S/W | 56.7% | 76.2±2.2% | 67.6±1.1% | 70.6±1.2% | 72.1±1.2% |
| D/A/NW | 0.4% | 66.6±9.9% | 59.7±8.9% | 71.2±7.5% | 74.1±9.2% |
| D/A/W | 0.9% | 66.4±2.4% | 57.0±5.0% | 72.9±3.8% | 73.6±4.3% |
| D/S/NW | 1.8% | 64.8±2.1% | 55.9±3.1% | 70.2±4.3% | 71.2±2.9% |
| D/S/W | 8.3% | 66.2±2.9% | 55.3±1.9% | 70.9±3.8% | 72.4±3.5% |
| sample mean | - | **75.1±1.8%** | 66.3±1.0% | 70.6±1.3% | 71.9±1.2% |
| group mean | 12.5% | 71.2±1.1% | 62.4±1.3% | 70.8±0.7% | **72.3±1.0%** |
| group max | 56.7% | **80.0±2.5%** | 70.0±2.6% | 77.6±4.1% | 79.9±5.6% |
| group min | 0.4% | 61.4±3.4% | 52.0±3.6% | 65.7±1.8% | **66.8±2.1%** |
| disc | 56.3% | 18.6±3.2% | 18.1±4.3% | **11.9±5.0%** | 13.1±5.6% |

Table 9: Accuracies on test set for MIMIC dataset. Standard deviations are computed across 5 splits.

| type | ratio | Naive | NaiveB | ZafarCE | ParetoFair CE | ParetoFair BS |
|---|---|---|---|---|---|---|
| A/A/NW | 5.7% | 0.027±0.004 | 0.198±0.019 | 0.138±0.029 | 0.291±0.04 | 0.252±0.033 |
| A/A/W | 13.3% | 0.027±0.005 | 0.196±0.013 | 0.2±0.026 | 0.285±0.022 | 0.247±0.019 |
| A/S/NW | 12.9% | 0.055±0.007 | 0.311±0.017 | 0.364±0.036 | 0.383±0.018 | 0.363±0.029 |
| A/S/W | 56.7% | 0.049±0.003 | 0.283±0.005 | 0.449±0.014 | 0.355±0.014 | 0.338±0.017 |
| D/A/NW | 0.4% | 1.007±0.133 | 0.362±0.055 | 1.042±0.189 | 0.335±0.03 | 0.34±0.041 |
| D/A/W | 0.9% | 0.922±0.068 | 0.292±0.041 | 0.785±0.121 | 0.287±0.022 | 0.277±0.031 |
| D/S/NW | 1.8% | 1.085±0.019 | 0.417±0.018 | 1.032±0.041 | 0.371±0.033 | 0.365±0.03 |
| D/S/W | 8.3% | 1.043±0.021 | 0.419±0.023 | 0.857±0.044 | 0.379±0.032 | 0.367±0.035 |
| sample mean | - | **0.159±0.003** | 0.285±0.006 | 0.437±0.014 | 0.347±0.013 | 0.327±0.013 |
| group mean | 12.5% | 0.527±0.022 | **0.31±0.009** | 0.608±0.026 | 0.336±0.009 | 0.319±0.002 |
| group max | 56.7% | 1.098±0.028 | 0.431±0.015 | 1.122±0.078 | 0.402±0.017 | **0.394±0.017** |
| group min | 0.4% | **0.025±0.003** | 0.188±0.016 | 0.138±0.029 | 0.261±0.023 | 0.233±0.022 |
| disc | 56.3% | 1.073±0.03 | 0.243±0.025 | 0.985±0.087 | **0.141±0.017** | 0.161±0.025 |

Table 10: Brier Score (BS) on test set for MIMIC dataset. Standard deviations are computed across 5 splits.

| type | ratio | Naive | NaiveB | ZafarCE | ParetoFair CE | ParetoFair BS |
|------|-------|-------|--------|---------|---------------|---------------|
| A/A/NW | 5.7% | 0.068±0.005 | 0.329±0.027 | 2.803±0.659 | 0.459±0.051 | 0.409±0.043 |
| A/A/W | 13.3% | 0.067±0.007 | 0.322±0.017 | 4.079±0.601 | 0.45±0.031 | 0.4±0.027 |
| A/S/NW | 12.9% | 0.121±0.009 | 0.471±0.02 | 7.697±0.91 | 0.564±0.02 | 0.538±0.034 |
| A/S/W | 56.7% | 0.111±0.003 | 0.434±0.007 | 9.238±0.363 | 0.531±0.019 | 0.508±0.021 |
| D/A/NW | 0.4% | 1.573±0.169 | 0.567±0.078 | 21.617±3.25 | 0.515±0.038 | 0.523±0.052 |
| D/A/W | 0.9% | 1.363±0.112 | 0.459±0.05 | 17.284±2.502 | 0.46±0.023 | 0.445±0.033 |
| D/S/NW | 1.8% | 1.58±0.046 | 0.604±0.027 | 22.097±0.961 | 0.552±0.037 | 0.544±0.037 |
| D/S/W | 8.3% | 1.532±0.046 | 0.609±0.033 | 18.446±0.9 | 0.563±0.036 | 0.547±0.041 |
| sample mean | - | **0.266±0.005** | 0.437±0.007 | 9.113±0.4 | 0.522±0.018 | 0.495±0.017 |
| group mean | 12.5% | 0.802±0.032 | **0.474±0.009** | 12.908±0.233 | 0.512±0.013 | 0.489±0.004 |
| group max | 56.7% | 1.65±0.055 | 0.635±0.014 | 23.343±1.058 | 0.587±0.019 | **0.578±0.019** |
| group min | 0.4% | **0.065±0.005** | 0.313±0.021 | 2.803±0.659 | 0.426±0.031 | 0.388±0.032 |
| disc | 56.3% | 1.585±0.059 | 0.323±0.03 | 20.54±1.089 | **0.162±0.025** | 0.191±0.037 |

Table 11: Cross Entropy (CE) on test set for MIMIC dataset. Standard deviations are computed across 5 splits.

| type | ratio | Naive | NaiveB | ZafarCE | ParetoFair CE | ParetoFair BS |
|------|-------|-------|--------|---------|---------------|---------------|
| A/A/NW | 5.7% | 0.046±0.004 | 0.063±0.007 | 0.069±0.014 | 0.109±0.016 | 0.101±0.004 |
| A/A/W | 13.3% | 0.043±0.002 | 0.051±0.006 | 0.1±0.013 | 0.092±0.027 | 0.083±0.015 |
| A/S/NW | 12.9% | 0.068±0.007 | 0.033±0.007 | 0.182±0.018 | 0.045±0.017 | 0.038±0.01 |
| A/S/W | 56.7% | 0.065±0.002 | 0.027±0.006 | 0.225±0.007 | 0.052±0.025 | 0.041±0.016 |
| D/A/NW | 0.4% | 0.55±0.094 | 0.164±0.049 | 0.522±0.095 | 0.149±0.037 | 0.146±0.06 |
| D/A/W | 0.9% | 0.43±0.057 | 0.09±0.03 | 0.394±0.061 | 0.155±0.057 | 0.123±0.053 |
| D/S/NW | 1.8% | 0.57±0.017 | 0.068±0.022 | 0.516±0.021 | 0.09±0.029 | 0.07±0.02 |
| D/S/W | 8.3% | 0.535±0.023 | 0.048±0.012 | 0.429±0.022 | 0.061±0.027 | 0.052±0.026 |
| sample mean | - | 0.115±0.004 | **0.037±0.003** | 0.219±0.007 | 0.062±0.021 | 0.052±0.012 |
| group mean | 12.5% | 0.288±0.016 | **0.068±0.006** | 0.305±0.013 | 0.094±0.023 | 0.082±0.021 |
| group max | 56.7% | 0.593±0.028 | 0.166±0.047 | 0.561±0.04 | 0.172±0.05 | **0.156±0.05** |
| group min | 0.4% | 0.042±0.002 | **0.025±0.003** | 0.069±0.014 | 0.039±0.013 | 0.032±0.011 |
| disc | 56.3% | 0.552±0.028 | 0.141±0.045 | 0.492±0.045 | 0.133±0.043 | **0.125±0.044** |

Table 12: Expected calibration error (ECE) on test set for MIMIC dataset. Standard deviations are computed across 5 splits.

| type | ratio | Naive | NaiveB | ZafarCE | ParetoFair CE | ParetoFair BS |
|------|-------|-------|--------|---------|---------------|---------------|
| A/A/NW | 5.7% | 0.322±0.096 | 0.125±0.02 | 0.425±0.301 | 0.147±0.017 | 0.166±0.016 |
| A/A/W | 13.3% | 0.251±0.061 | 0.088±0.015 | 0.229±0.133 | 0.127±0.026 | 0.124±0.019 |
| A/S/NW | 12.9% | 0.223±0.038 | 0.058±0.013 | 0.323±0.198 | 0.077±0.03 | 0.07±0.013 |
| A/S/W | 56.7% | 0.203±0.007 | 0.046±0.011 | 0.496±0.184 | 0.091±0.034 | 0.069±0.02 |
| D/A/NW | 0.4% | 0.828±0.115 | 0.341±0.094 | 0.522±0.095 | 0.293±0.048 | 0.422±0.266 |
| D/A/W | 0.9% | 0.879±0.07 | 0.183±0.074 | 0.509±0.201 | 0.206±0.033 | 0.191±0.072 |
| D/S/NW | 1.8% | 0.822±0.052 | 0.158±0.082 | 0.553±0.067 | 0.152±0.027 | 0.129±0.029 |
| D/S/W | 8.3% | 0.865±0.027 | 0.11±0.038 | 0.563±0.182 | 0.136±0.044 | 0.112±0.041 |
| sample mean | - | 0.294±0.013 | **0.067±0.007** | 0.441±0.113 | 0.104±0.029 | 0.089±0.015 |
| group mean | 12.5% | 0.549±0.031 | **0.139±0.007** | 0.453±0.094 | 0.154±0.024 | 0.16±0.047 |
| group max | 56.7% | 0.914±0.045 | 0.364±0.061 | 0.7±0.105 | **0.293±0.048** | 0.429±0.259 |
| group min | 0.4% | 0.196±0.007 | **0.042±0.008** | 0.146±0.04 | 0.071±0.025 | 0.059±0.013 |
| disc | 56.3% | 0.719±0.044 | 0.322±0.065 | 0.554±0.097 | **0.222±0.062** | 0.37±0.262 |

Table 13: Maximum calibration error (MCE) on test set for MIMIC dataset. Standard deviations are computed across 5 splits.

| Groups | ratio | ParetoFair | NaiveB | Naive | ParetoFair | NaiveB | Naive |
|---|---|---|---|---|---|---|---|
| akiec | 2.5% | 51.9% | 55.6% | 3.7% | 55.4% | 66.3% | 55.3% |
| bcc | 2.7% | 56.7% | 76.7% | 56.7% | 57.6% | 77.5% | 66.7% |
| bkl | 7.2% | 59.5% | 58.2% | 36.7% | 64.5% | 71.6% | 64.3% |
| df | 0.5% | 66.7% | 33.3% | 0.0% | 59.8% | 68.1% | 81.6% |
| nv | 81.0% | 83.5% | 90.9% | 96.0% | 81.7% | 89.1% | 93.7% |
| vasc | 1.3% | 71.4% | 85.7% | 0.0% | 66.9% | 74.7% | 76.1% |
| mel | 4.8% | 53.8% | 48.1% | 32.7% | 63.6% | 32.1% | 63.6% |
| sample mean | - | 78.6% | **84.8%** | 83.6% | 78.0% | 85.6% | 88.1% |
| group mean | 14.3% | 63.4% | **64.1%** | 32.3% | 64.2% | 73.6% | 71.6% |
| group max | 81.0% | 83.5% | 90.9% | **96.0%** | 81.7% | 89.1% | 93.7% |
| group min | 0.5% | **51.9%** | 33.3% | 0.0% | 55.4% | 66.3% | 55.3% |
| disc | 80.4% | **31.7%** | 57.5% | 96.0% | 26.3% | 22.8% | 38.4% |

(a) Accuracy          (b) Confidence

Table 14: Accuracies (left) and confidence (right) on test set for HAM10000 dataset.

| type | ratio | ParetoFair | NaiveB | Naive | ParetoFair | NaiveB | Naive |
|---|---|---|---|---|---|---|---|
| akiec | 2.5% | 0.741 | 0.671 | 1.289 | 1.547 | 1.638 | 3.879 |
| bcc | 2.7% | 0.549 | 0.341 | 0.613 | 1.095 | 0.678 | 1.172 |
| bkl | 7.2% | 0.62 | 0.606 | 0.931 | 1.345 | 1.206 | 2.258 |
| df | 0.5% | 0.536 | 0.898 | 1.721 | 0.989 | 1.674 | 6.486 |
| nv | 81.0% | 0.241 | 0.128 | 0.054 | 0.58 | 0.28 | 0.115 |
| vasc | 1.3% | 0.36 | 0.246 | 1.604 | 0.747 | 0.499 | 4.564 |
| mel | 4.8% | 0.586 | 0.657 | 0.941 | 1.213 | 1.28 | 2.137 |
| sample mean | - | 0.308 | **0.213** | 0.234 | 0.708 | **0.449** | 0.579 |
| group mean | 14.3% | 0.519 | **0.507** | 1.022 | 1.074 | **1.037** | 2.944 |
| group max | 81.0% | **0.741** | 0.898 | 1.721 | **1.547** | 1.674 | 6.486 |
| group min | 0.5% | 0.241 | 0.128 | **0.054** | 0.58 | 0.28 | **0.115** |
| disc | 80.4% | **0.501** | 0.769 | 1.667 | **0.967** | 1.394 | 6.371 |

(a) Brier Score          (b) Cross Entropy

Table 16: Brier Score (left) and Cross Entropy (right) on test set for HAM10000 dataset.

| type | ratio | ParetoFair | NaiveB | Naive | ParetoFair | NaiveB | Naive |
|---|---|---|---|---|---|---|---|
| akiec | 2.5% | 0.249 | 0.254 | 0.516 | 0.754 | 0.447 | 0.966 |
| bcc | 2.7% | 0.147 | 0.134 | 0.161 | 0.269 | 0.408 | 0.418 |
| bkl | 7.2% | 0.171 | 0.156 | 0.304 | 0.731 | 0.294 | 0.744 |
| df | 0.5% | 0.452 | 0.57 | 0.816 | 0.718 | 0.853 | 0.985 |
| nv | 81.0% | 0.025 | 0.034 | 0.032 | 0.143 | 0.248 | 0.273 |
| vasc | 1.3% | 0.256 | 0.202 | 0.761 | 0.557 | 0.539 | 0.956 |
| mel | 4.8% | 0.14 | 0.212 | 0.323 | 0.277 | 0.289 | 0.836 |
| sample mean | - | **0.055** | 0.065 | 0.095 | **0.219** | 0.27 | 0.367 |
| group mean | 14.3% | **0.206** | 0.223 | 0.416 | 0.493 | **0.44** | 0.74 |
| group max | 81.0% | **0.452** | 0.57 | 0.816 | **0.754** | 0.853 | 0.985 |
| group min | 0.5% | **0.025** | 0.034 | 0.032 | **0.143** | 0.248 | 0.273 |
| disc | 80.4% | **0.426** | 0.535 | 0.784 | 0.611 | **0.605** | 0.712 |

(a) ECE          (b) MCE

Table 18: Expected Calibration Error (left) and Maximum Calibration Error (right) on test set for HAM10000 dataset.

| type | ratio | Naive | NaiveB | Feldman | Kamishima | Zafar | ParetoFair LR |
|---|---|---|---|---|---|---|---|
| Male | 67.9% | 92.3±0.4% | 92.3±0.3% | 92.4±0.3% | 92.6±0.4% | 92.2±0.7% | 92.4±0.3% |
| Female | 32.1% | 80.5±0.4% | 80.3±0.7% | 80.8±0.3% | 80.9±0.4% | 80.8±0.1% | 81.0±0.5% |
| sample mean | - | 84.3±0.3% | 84.2±0.5% | 84.5±0.3% | 84.6±0.3% | 84.4±0.3% | 84.7±0.3% |
| group mean | 50.0% | 86.4±0.2% | 86.3±0.4% | 86.6±0.3% | 86.7±0.2% | 86.5±0.4% | 86.7±0.2% |
| disc | 35.9% | 11.9±0.7% | 12.0±0.7% | 11.6±0.2% | 11.7±0.7% | 11.4±0.7% | 11.4±0.6% |

| type | ratio | Naive | NaiveB | Feldman | Kamishima | Zafar | ParetoFair LR |
|---|---|---|---|---|---|---|---|
| White Male | 39.7% | 90.8±0.3% | 90.8±0.3% | 90.6±0.3% | 90.4±1.1% | 90.5±0.6% | 90.7±0.3% |
| Other | 60.3% | 79.8±0.5% | 79.7±0.7% | 80.4±0.6% | 79.4±2.3% | 80.1±0.1% | 80.6±0.6% |
| sample mean | - | 84.1±0.2% | 84.1±0.4% | 84.5±0.3% | 83.8±1.8% | 84.2±0.2% | 84.6±0.4% |
| group mean | 50.0% | 85.3±0.1% | 85.2±0.3% | 85.5±0.3% | 84.9±1.6% | 85.3±0.3% | 85.6±0.4% |
| disc | 20.5% | 11.0±0.8% | 11.1±0.8% | 10.2±0.7% | 11.0±1.4% | 10.3±0.6% | 10.2±0.6% |

Table 20: Accuracies on test set for Adult dataset. Standard deviations are computed across 5 splits.

| type | ratio | Naive | NaiveB | Feld | Kam | Zafar | ParetoFair LR |
|---|---|---|---|---|---|---|---|
| Male | 32.1% | 0.116±0.004 | 0.117±0.005 | 0.115±0.004 | 0.11±0.005 | 0.115±0.013 | 0.114±0.003 |
| Female | 67.9% | 0.268±0.004 | 0.272±0.007 | 0.263±0.005 | 0.258±0.003 | 0.26±0.003 | 0.261±0.006 |
| sample mean | - | 0.22±0.003 | 0.222±0.006 | 0.215±0.004 | 0.21±0.003 | 0.214±0.006 | 0.214±0.004 |
| group mean | 50.0% | 0.192±0.003 | 0.194±0.006 | 0.189±0.003 | 0.184±0.003 | 0.188±0.008 | 0.188±0.003 |
| disc | 35.9% | 0.152±0.005 | 0.155±0.005 | 0.148±0.005 | 0.147±0.005 | 0.144±0.011 | 0.147±0.007 |

| type | ratio | Naive | NaiveB | Feld | Kam | Zafar | ParetoFair LR |
|---|---|---|---|---|---|---|---|
| White Male | 39.7% | 0.139±0.003 | 0.137±0.004 | 0.135±0.004 | 0.14±0.02 | 0.138±0.013 | 0.133±0.004 |
| Other | 60.3% | 0.28±0.004 | 0.28±0.008 | 0.271±0.005 | 0.284±0.038 | 0.269±0.002 | 0.267±0.007 |
| sample mean | - | 0.224±0.002 | 0.223±0.005 | 0.217±0.003 | 0.227±0.031 | 0.217±0.006 | 0.214±0.005 |
| group mean | 50.0% | 0.21±0.002 | 0.208±0.005 | 0.203±0.003 | 0.212±0.029 | 0.204±0.007 | 0.2±0.005 |
| disc | 20.5% | 0.142±0.007 | 0.143±0.007 | 0.136±0.007 | 0.144±0.019 | 0.131±0.012 | 0.133±0.008 |

Table 21: Brier Score (BS) on test set for Adult dataset. Standard deviations are computed across 5 splits.

| type | ratio | Naive | NaiveB | Feld | Kam | Zafar | ParetoFair LR |
|---|---|---|---|---|---|---|---|
| Male | 67.9% | 0.204±0.009 | 0.204±0.011 | 0.2±0.005 | 0.189±0.006 | 0.201±0.025 | 0.207±0.017 |
| Female | 32.1% | 0.411±0.006 | 0.416±0.011 | 0.402±0.007 | 0.395±0.004 | 0.406±0.01 | 0.41±0.014 |
| sample mean | - | 0.345±0.006 | 0.348±0.011 | 0.337±0.005 | 0.329±0.004 | 0.34±0.01 | 0.345±0.013 |
| group mean | 50.0% | 0.308±0.007 | 0.31±0.011 | 0.301±0.005 | 0.292±0.004 | 0.303±0.013 | 0.309±0.013 |
| disc | 35.9% | 0.207±0.007 | 0.211±0.005 | 0.203±0.008 | 0.206±0.007 | 0.206±0.027 | 0.202±0.016 |

| type | ratio | Naive | NaiveB | Feld | Kam | Zafar | ParetoFair LR |
|---|---|---|---|---|---|---|---|
| White Male | 39.7% | 0.238±0.005 | 0.233±0.008 | 0.227±0.006 | 0.254±0.075 | 0.237±0.022 | 0.233±0.013 |
| Other | 60.3% | 0.428±0.007 | 0.428±0.011 | 0.413±0.007 | 0.463±0.116 | 0.418±0.007 | 0.415±0.015 |
| sample mean | - | 0.352±0.004 | 0.35±0.009 | 0.339±0.005 | 0.38±0.099 | 0.346±0.009 | 0.343±0.013 |
| group mean | 50.0% | 0.333±0.003 | 0.33±0.008 | 0.32±0.004 | 0.359±0.095 | 0.327±0.011 | 0.324±0.013 |
| disc | 20.5% | 0.19±0.009 | 0.194±0.009 | 0.187±0.01 | 0.209±0.042 | 0.181±0.024 | 0.183±0.013 |

Table 22: Cross Entropy (CE) on test set for Adult dataset. Standard deviations are computed across 5 splits.

| type | ratio | Naive | NaiveB | Feld | Kam | Zafar | ParetoFair LR |
|---|---|---|---|---|---|---|---|
| Male | 67.9% | 0.026±0.008 | 0.023±0.007 | 0.024±0.003 | 0.011±0.003 | 0.021±0.023 | 0.017±0.007 |
| Female | 32.1% | 0.013±0.004 | 0.014±0.005 | 0.01±0.004 | 0.012±0.002 | 0.014±0.005 | 0.013±0.004 |
| sample mean | - | 0.017±0.005 | 0.017±0.005 | 0.014±0.003 | 0.011±0.002 | 0.016±0.011 | 0.014±0.003 |
| group mean | 50.0% | 0.019±0.006 | 0.019±0.005 | 0.017±0.002 | 0.011±0.002 | 0.018±0.014 | 0.015±0.003 |
| disc | 35.9% | 0.013±0.004 | 0.01±0.006 | 0.014±0.006 | 0.003±0.002 | 0.011±0.016 | 0.009±0.005 |

| type | ratio | Naive | NaiveB | Feld | Kam | Zafar | ParetoFair LR |
|---|---|---|---|---|---|---|---|
| White Male | 39.7% | 0.026±0.006 | 0.024±0.008 | 0.019±0.002 | 0.016±0.012 | 0.026±0.02 | 0.019±0.006 |
| Other | 60.3% | 0.018±0.005 | 0.016±0.004 | 0.014±0.005 | 0.021±0.022 | 0.019±0.008 | 0.015±0.004 |
| sample mean | - | 0.021±0.004 | 0.019±0.006 | 0.016±0.003 | 0.019±0.018 | 0.022±0.013 | 0.017±0.002 |
| group mean | 50.0% | 0.022±0.004 | 0.02±0.006 | 0.016±0.003 | 0.019±0.017 | 0.023±0.014 | 0.017±0.003 |
| disc | 20.5% | 0.008±0.005 | 0.007±0.005 | 0.007±0.003 | 0.006±0.01 | 0.01±0.009 | 0.008±0.005 |

Table 23: Expected calibration error (ECE) on test set for Adult dataset. Standard deviations are computed across 5 splits.

| type | ratio | Naive | NaiveB | Feld | Kam | Zafar | ParetoFair LR |
|---|---|---|---|---|---|---|---|
| Male | 67.9% | 0.064±0.012 | 0.065±0.034 | 0.064±0.017 | 0.072±0.01 | 0.077±0.03 | 0.062±0.022 |
| Female | 32.1% | 0.027±0.013 | 0.031±0.009 | 0.03±0.01 | 0.031±0.008 | 0.035±0.008 | 0.024±0.008 |
| sample mean | - | 0.039±0.009 | 0.042±0.016 | 0.041±0.006 | 0.044±0.008 | 0.049±0.011 | 0.036±0.01 |
| group mean | 50.0% | 0.045±0.008 | 0.048±0.021 | 0.047±0.008 | 0.052±0.009 | 0.056±0.016 | 0.043±0.013 |
| disc | 35.9% | 0.037±0.019 | 0.034±0.027 | 0.034±0.024 | 0.041±0.004 | 0.042±0.03 | 0.038±0.02 |

| type | ratio | Naive | NaiveB | Feld | Kam | Zafar | ParetoFair LR |
|---|---|---|---|---|---|---|---|
| White Male | 39.7% | 0.043±0.018 | 0.053±0.015 | 0.045±0.008 | 0.061±0.019 | 0.08±0.033 | 0.052±0.014 |
| Other | 60.3% | 0.05±0.011 | 0.035±0.007 | 0.035±0.019 | 0.039±0.034 | 0.044±0.01 | 0.032±0.011 |
| sample mean | - | 0.047±0.011 | 0.042±0.01 | 0.039±0.011 | 0.048±0.025 | 0.058±0.017 | 0.04±0.009 |
| group mean | 50.0% | 0.047±0.011 | 0.044±0.011 | 0.04±0.009 | 0.05±0.023 | 0.062±0.02 | 0.042±0.009 |
| disc | 20.5% | 0.016±0.014 | 0.017±0.009 | 0.023±0.009 | 0.032±0.02 | 0.037±0.029 | 0.023±0.014 |

Table 24: Maximum calibration error (MCE) on test set for Adult dataset. Standard deviations are computed across 5 splits.

### A.8 ANALYSIS OF PARETO-OPTIMAL CLASSIFIERS IN THE INFINITE DATA AND MODEL CAPACITY REGIME

In this section we analyze the form of Pareto-optimal solutions to classification (and regression) tasks in the asymptotically ideal case where we have infinite capacity hypothesis classes, that is, our hypothesis class $\mathcal{H}$ contains every function mapping points from the observation space $\mathcal{X}$ to the classification or regression space $\mathbb{R}^{|\mathcal{Y}|}$.

We additionally assume that the joint distributions between target variables $Y$ and observation variables $X$ given every sensitive attribute are known (i.e., $P(Y, X|A)$ is known), and that the loss function $L(h(x), y)$ is convex with respect to $h(x)$.

Any joint loss of the form

$$\phi_{P_A}(\{R_a(h)\}) = \mathbb{E}_{X,Y,A}[L(h(X), Y] = \sum_{a \in \mathcal{A}} P_a R_a(h), \tag{3}$$

produces solutions which are already in the Pareto front of $\{R_a(h)\}$ for any distribution of $A \sim P_A$. Since the loss function is convex with respect to $h(X)$ and the hypothesis class is complete, it can also be shown that for any point in the Pareto front, there exists a value of $P_A$ such that that point is reached by minimizing $\phi_{P_A}(\{R_a(h)\})$ Geoffrion (1968); Koski (1985); Miettinen (2012)

We can therefore analyze the Bayes-optimal classifier $h_{P_A}$ which minimizes the Naive risk $\phi_{P_A}(\{R_a(h)\})$, and analytically compute the sub-population risks $R_a(h_{P_A})$ induced. In general, we can write

$$
\begin{aligned}
\mathbb{E}[Y|x] \quad &= \sum_{a \in \mathcal{A}} \mathbb{E}[Y|x, a] P(a|x), \\
&= \frac{\sum_{a \in \mathcal{A}} \mathbb{E}[Y|x,a] P(x|a) P_a}{\sum_{a \in \mathcal{A}} P(x|a) P_a}, \\
h_{P_A}(X) \quad &= \arg\min_{\eta} \mathbb{E}[L(\eta, Y)|X], \\
R_a(h_{P_A}) \quad &= \mathbb{E}_{X,Y}[L(h_{P_A}(X), Y)|A = a],
\end{aligned}
\tag{4}
$$

and in the particular case where target variable $Y$ is categorical, and the classifier loss is an L2 loss against the one-hot encoding of variable $Y$ the equations reduce to

$$
\begin{aligned}
\mathbb{E}[Y|x] \quad &= \frac{\sum_{a \in \mathcal{A}} \vec{P}[Y|x,a] P(x|a) P_a}{\sum_{a \in \mathcal{A}} P(x|a) P_a}, \\
h_{P_A}(X) \quad &= \mathbb{E}[Y|x], \\
R_a(h_{P_A}) \quad &= \mathbb{E}_X[\sum_y P(y|a, X) \sum_{y'} (h_{P_A}^{y'}(X) - \delta[y - y'])^2].
\end{aligned}
\tag{5}
$$

