# OpenReview forum: "Pareto Optimality in No-Harm Fairness"
_ICLR.cc/2020/Conference — Reject_

### Official Review · AnonReviewer3 · 2019-10-23
**Official Blind Review #3**

**Rating:** 3

**Review:**

This paper introduces a new kind of algorithmic fairness framework where the focus is on first finding a fair classifier that does "no harm" and then in a subsequent step potentially allow doing harm in order to achieve even fairer outcomes. Fairness is here understood as risk disparity: how different are the risks achieved by our model in the various subgroups. The risk is task-dependent and can be something like a cross-entropy loss for classification problems. The goal is to have similar risks in the subgroups that correspond to sensitive attributes.

It is often impossible to have equal risks without doing some harm because some subgroups might have higher noise-levels or fewer samples so that it is fundamentally not possible to achieve low risks in these subgroups. The only way then to make risks equal is by *increasing* the risk in all the other subgroups. This is not always desirable, so this paper presents a method for finding a model that is as fair as possible without doing harm.

---

The basic idea of this paper is solid, but the mathematical definitions don't seem to capture that idea. Definitions 3.1, 3.2 and 3.3 define together the "optimal Pareto-fair classifier". But this doesn't seem to correspond to what was described before.

Here is an example to show what I mean:

Say, we have two subgroups: $a=0$ and $a=1$ and the risk is a binary classification loss. Furthermore, we have two classifiers $h_1$ and $h_2$. Now, say the achieved risk is such that $h_1$ achieves 80% accuracy on $a=0$ and 30% accuracy on $a=1$ (to make it precise it should be classification loss instead of accuracy but those two should be basically equivalent); classifier $h_2$ achieves 60% on $a=0$ and 31% on $a=1$.

According to definition 3.1, neither of them dominates the other. So they could both be in the Pareto front. But classifier $h_2$ clearly does harm to $a=0$. And furthermore, definition 3.3 will choose $h_2$ over $h_1$ as the optimal classifier as the gap is smaller with $h_2$. So how can you claim that the optimal Pareto-fair classifier does no harm?

Now, the classifier that you train in the end is actually from a much smaller subset that happens to be in the Pareto front: it is one that minimizes the overall risk (Lemma 3.2) and this subset might really do no harm, but that is still not obvious to me.

Another problem that I see is that the proof for Lemma 3.1 is not constructive, so while there might exist a classifier $h_p$ that dominates $h$, you might not be able to find it; and just using $h$ might turn out to be a reasonable choice.

Minor comments:

- the plots don't seem to be vector graphics

**Experience Assessment:**

I have published one or two papers in this area.

**Review Assessment: Checking Correctness Of Derivations And Theory:**

I assessed the sensibility of the derivations and theory.

**Review Assessment: Checking Correctness Of Experiments:**

I assessed the sensibility of the experiments.

**Review Assessment: Thoroughness In Paper Reading:**

I made a quick assessment of this paper.

---

> ### Author Response · Authors · 2019-11-15
> **Response to Reviewer 3**
>
> We thank the reviewer for their comments and analysis. We are glad he/she agrees that this is a worthwhile problem to look into, and found our formulation interesting. We improved the overall presentation considering your comments and suggestions.
>
> We apologize for the lack of clarity in some of the definitions and nomenclature. The example you describe is correct, the second classifier could very well be Pareto efficient and it exhibits a smaller risk disparity gap, so it would be our choice of Pareto-fair classifier.
>
> In general, all Pareto-efficient classifiers require some performance tradeoff between subgroups, which is why we changed the naming of our classifier to “Pareto-fair classifier”, and we will change the title to “Pareto optimality in no-unnecessary-harm fairness” to better reflect the reality of what our method is proposing. Overall, while the example you provide is certainly possible, empirical results show much more reasonable and desireable tradeoffs between the best and worst performing subgroups. For example, compared against the baseline classifier, the accuracy of the worst performing subgroup in the mortality prediction task (MIMIC) increased by 50%, while the best performing subgroup had a performance decrease of 12%. Please note also that our framework doesn’t need to have knowledge of the sensitive variable, which is very important in many applications.
>
>
> We reworked the Problem Statement and Optimization method section to give a clearer overview of the proposed algorithm to be more constructive. In particular, we added an extra Lemma in Supplementary Material  (Lemma A.5) that proves that, under some hypothesis, our algorithm is effectively converging to the Pareto-fair classifier.  A more general proof under milder assumptions is something we are actively pursuing. We significantly expanded the results section, both in the main paper and in supplementary material, to give a broader overview on the advantages and disadvantages of our framework compared to other methodologies.
>
> We increased the overall definition of the plots and enlarged some of the figures. Additional explanations on the figures were also provided in the paper

---

### Official Review · AnonReviewer1 · 2019-10-28
**Official Blind Review #1**

**Rating:** 3

**Review:**

# 1403

General

The paper proposes a method to achieve the no-harm fair model that will lie on the Pareto-optimal front, but has the minimum risk discrimination gap. While the problem formulation is interesting, the paper is not very easy to follow and there are some aspects that the paper needs to get improved to get published, in my opinion.

Con & Questions:
The notations for the mathematical expressions are not very clearly explained. I think it causes unnecessary confusions.
Deferring the main algorithm to the Supplementary seems weird. I think the algorithm pseudo-code should be included in the main paper so that the full method can be appreciated properly.
The quality of the figures is very poor. Fonts are too small and hard to read.
Table 1 is  confusing. The numbers it presenting is not clearly described. In the caption, it says they are cross-entropy risk / accuracies, but the caption also says Pareto-Fair achieves the lowest mean risk and risk disparity. Where is the information about the risk disparity? What exactly are the number for the row Discrimination?

**Experience Assessment:**

I have read many papers in this area.

**Review Assessment: Checking Correctness Of Derivations And Theory:**

I assessed the sensibility of the derivations and theory.

**Review Assessment: Checking Correctness Of Experiments:**

I assessed the sensibility of the experiments.

**Review Assessment: Thoroughness In Paper Reading:**

I read the paper at least twice and used my best judgement in assessing the paper.

---

> ### Author Response · Authors · 2019-11-15
> **Response to Reviewer 1**
>
> We thank the reviewer for their constructive comments. We are glad he/she agrees that this is a worthwhile problem to look into, and found our formulation interesting. We improved the overall presentation considering your comments and suggestions, we hope readability is improved. We included theoretical results and analysis on the proposed algorithm, and expanded the results section.
>
> We reorganized the Problem Statement and Optimization Method section to be easier to read and give a clearer overview of the proposed algorithm.
>
> We agree that having the main algorithm on the Supplementary Material section was not helping the readability of the overall paper, so we moved it to the main paper under the Optimization Method  Section.
>
> Figures were overall enlarged, and caption font size was also increased. We provided a better explanation on the figures in the main text.
>
> We made the description for the metrics presented in the paper more clear in the supplementary material, and we clarified that all metrics are computed per subgroup. We explicitly added rows to show the best and worst performing subgroup for every metric, and clarified that the Discrimination (disc) row corresponds to the difference between the best and worst performing subgroup for each metric.  The experiments have been  improved  overall and show more clearly the advantages of the proposed framework.

---

### Official Review · AnonReviewer2 · 2019-10-30
**Official Blind Review #2**

**Rating:** 3

**Review:**

This paper considers the notion of "no-harm" group fairness, i.e. trying to reduce the risk gap between minority and majority groups without excessive reduction in performance on the majority groups. Authors formalize the problem by defining a Pareto fair classifier, i.e. one that minimizes the risk gaps between groups and belongs to the family of Pareto classifiers containing the classifier minimizing the empirical risk. Authors suggest an optimization procedure for finding the Pareto fair classifier and demonstrate its performance on multiple datasets.

Pros:
I think that studying "no-harm" classifiers is an important topic given the alarming tendency of some of the recent group fairness approaches to achieve fairness by essentially driving down the performance on the majority group without improving on the minority group. Decision making in medical applications is one of the prominent examples where "no-harm" is absolutely needed, as authors suggested. The mathematical formulation of the problem around the notion of Pareto optimality also seems reasonable.

Cons:
My concerns are related to counter-intuitive experimental results and lack of clarity in parts of the presentation.

Figure 1 seems important for understanding the ideas in the paper, but is not explained in much detail. Analogous to it Figure 2 is lacking important details. In the upper left plot, what are the decision boundaries of the baselines? What are the baselines risks in the center top figure, particularly for the equal risk classifier? It is hard to see from the right figure if the proposed classifier achieves the "no-harm" fairness over the equal risk classifier - numerical summary in a table could help. Finally, why is it necessary to use a neural network (which seems to be the case based on the supplement A.3) for the toy problem? I would recommend working through a toy example in more detail using a linear classifier to verify the correctness of the proposed technique and improve the overall clarity. Further, absence of a toy problem with linear classifier is alarming given there is not much discussion of the algorithm and its convergence properties.

Regarding the real data experiments, none seem to showcase the "no-harm" versus "zero-gap" fairness tradeoff motivating this paper.

MIMIC-III results seem to contradict the main story of the paper. The minority group appears to be "D/A/NW", then there should be a "harming" group fairness classifier achieving close to 0 discrimination at the cost of lowering performance on other subgroups. The "no-harm" classifier should then achieve a similar or slightly lower performance on "D/A/NW", but much better results on other sub-groups. Despite, the "no-harm" classifiers seems to outperform other approaches on "D/A/NW" by a good margin. Next, it appears that "Naive+Zafar" (it also would be helpful to have a brief discussion of the baselines considered) approach was not configured to eliminate A/S disparity as suggested by poor results on the D/A/NW and D/A/W, while it performs very well on other subpopulations.

Skin Lesion classification experiment departs from the problem of fairness and considers the problem of classification with unbalanced classes instead. Results are again counterintuitive. "Rebalanced Naive" only mildly improves over the "Naive" approach, while proposed algorithm seems to achieve a quite significant mean accuracy improvement. This again does not show the motivating "harm" vs "zero gap" tradeoff, but it could be interesting as the imbalanced classification is an important problem by itself. Could you please compare to more advanced imbalanced classification algorithms?

Results on the Adult and German Credit datasets are very similar across competing methods.

Acknowledgement and references to the individual fairness line of works are missing when presenting the problem of fairness in machine learning.
Font size in tables and Figure 2 legends is too small.
Typo in the last sentence on page 8: "have highly impactful" -> "are highly impactful".

**Experience Assessment:**

I have published one or two papers in this area.

**Review Assessment: Checking Correctness Of Derivations And Theory:**

I assessed the sensibility of the derivations and theory.

**Review Assessment: Checking Correctness Of Experiments:**

I carefully checked the experiments.

**Review Assessment: Thoroughness In Paper Reading:**

I read the paper at least twice and used my best judgement in assessing the paper.

---

> ### Author Response · Authors · 2019-11-15
> **Response to Reviewer 2**
>
> We thank the reviewer for their insightful comments that motivated us to improve our work, and we are glad the reviewer found the topic relevant. We have significantly improved the overall presentation (as detailed next), provided a new proof further supporting not just the framework but the computational component, and new experimental results. We believe the reviewer’s inputs are all addressed now and have helped us to improve the paper. We detail the key changes next.
>
> We agree that Figure 1 was not sufficiently explained, to that end we added a better explanation on what each of the plots represent and how they relate to the definitions presented in the paper.
>
> Likewise, the axes on figure 2 have been made clearer, and the figure has been enlarged overall. We also added additional explanations on why linear logistic regression is insufficient to recover the optimal Pareto-fair classifier in this scenario.
>
> We reworked the Problem Statement and Optimization Method section to give a clearer overview of the proposed algorithm, and derived conditions on which the algorithm converges to the optimal solution, supporting our design choices.
>
> We carefully ran the experiments on MIMIC-III and now show a clear advantage on reducing accuracy disparity against Zafar and the Naive baselines. The performance disparity is reduced further when we apply Hardt post-processing, but at a steep cost in accuracy for both the advantaged and disadvantaged classes.
>
> We added a more thorough explanation of the baselines in supplementary material. In particular, we explain that Zafar on MIMIC was trained to control for age and ethnicity, but not for outcome, since Zafar attempts to decorrelate sensitive attributes and classification boundaries.
>
> Skin lesion classification experiment now show a clear discrimination advantage of the Pareto-fair classifier compared to both Naive and Balanced Naive, along with better Expected Calibration Errors. The Naive classifier has very similar performance to the Balanced Naive classifier on this dataset, with the exception of the minority class, which occurs .5% of the time, on which the performance difference is severe. Unfortunately, we were unable to produce comparisons against other unbalanced classification algorithms in time. We could not find and run an appropriate algorithm before the deadline. We will include this comparison in the next version.
>
> We added references to works in Individual Fairness in the Introduction.
>
> Typos were addressed

---

### Decision · Program_Chairs · 2019-12-19

**Decision:**

Reject

**Comment:**

This manuscript outlines procedures to address fairness as measured by disparity in risk across groups. The manuscript is primarily motivated by methods that can achieve "no-harm" fairness, i.e., achieving fairness without increasing the risk in subgroups.

The reviewers and AC agree that the problem studied is timely and interesting. However, in reviews and discussion, the reviewers noted issues with clarity of the presentation, and sufficient justification of the results. The consensus was that the manuscript in its current state is borderline, and would have to be significantly improved in terms of clarity of the discussion, and possibly improved methods that result in more convincing results.